# RODEO: ROBUST OUT-OF-DISTRIBUTION DETECTION VIA EXPOSING ADAPTIVE OUTLIERS

## ABSTRACT

Detecting out-of-distribution (OOD) input samples at the time of inference is a key element in the trustworthy deployment of intelligent models. While there has been tremendous improvement in various variants of OOD detection in recent years, detection performance under adversarial settings lags far behind the performance in the standard setting. In order to bridge this gap, we introduce RODEO, a data-centric approach that generates effective outliers for robust OOD detection. More specifically, we first show that targeting the classification of adversarially perturbed in- and out-of-distribution samples through outlier exposure (OE) could be an effective strategy for the mentioned purpose, as long as the training outliers meet certain quality standards. We hypothesize that the outliers in the OE technique should possess several characteristics simultaneously to be effective in the adversarial training: diversity, and both conceptual differentiability and analogy to the inlier samples. These aspects seem to play a more critical role in the adversarial setup compared to the standard training. propose an adaptive OE method to generate near-distribution and diverse outliers by incorporating both text and image domain information. This process helps satisfy the mentioned criteria for the generated outliers and significantly enhances the performance of the OE technique, particularly in adversarial settings. Our method demonstrates its effectiveness across various detection setups, such as novelty detection (ND), Open-Set Recognition (OSR), and OOD detection. Furthermore, we conduct a comprehensive comparison of our approach with other OE techniques in adversarial settings to showcase its effectiveness.

A widespread assumption in model development in machine learning is that the test data is drawn from the same distribution as the training data, known as the closed-set assumption. However, during inference, models may encounter a wide variety of anomalous data, significantly deviating from the in-distribution (1; 2; 3). This phenomenon creates issues in safety-critical applications, which require the detection and distinct treatment of outliers (4). Detection of such inputs has various flavors, such as ND, OOD detection, and OSR, and several techniques have been developed for each of these problem setups (3; 5; 6).

Robustness against adversarial attacks is another important ML safety problem in real-world model development. Adversarial attacks are the imperceptible input perturbations that are designed to cause the model to make incorrect predictions (7; 8; 9). Despite the emergence of many promising outlier detection methods in recent years (10; 11; 12), they often suffer significant performance drops when subjected to adversarial attacks, which aim to convert inliers into outliers and vice versa. In light of this, recently, several robust outlier detection methods have been proposed (13; 14; 15; 16; 17; 18; 19; 20). However, their results are still unsatisfactory, sometimes performing even worse than random detection, and are often focused on simplified cases of outlier detection rather than being broadly applicable. Motivated by this, we aim to provide a robust and unified solution for outlier detection that can perform well in both clean and adversarial settings.

Adversarial training, which is the augmentation of the training samples with adversarial perturbations, is among the best practices for making the models robust. However, this approach is less effective in outlier detection, as OOD patterns are unknown during training, thus preventing the training of models with adversarial perturbations associated with these outliers. For this reason, recent robust outlier detection methods use the Outlier Exposure (OE) technique (21) in combination with adversarial training to tackle this issue (13; 15; 20). In OE, the auxiliary outlier samples are typically obtained

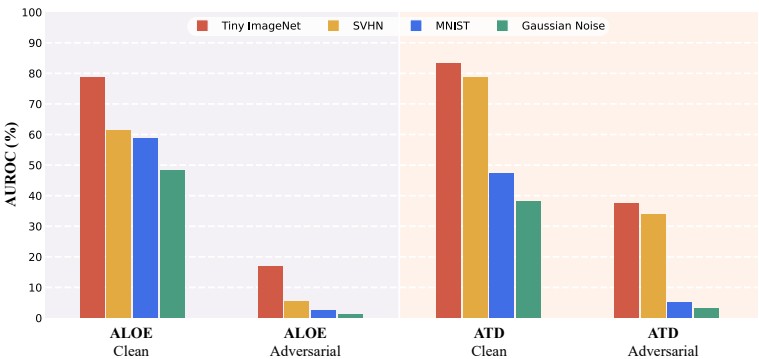

Figure 1: In this experiment, we aimed to reproduce the ALOE and ATD methods for OOD detection on the CIFAR10 vs CIFAR100 task. The original implementations of ALOE and ATD utilize the Tiny ImageNet dataset as an auxiliary outlier dataset during training to improve robust detection. Without changing any other component, we replaced the Tiny ImageNet auxiliary outlier set with SVHN, MNIST, and Gaussian noise and repeated the experiments. This replacement significantly degraded the OOD detection performance of ALOE and ATD on the cifar10 vs cifar100 task, especially under adversarial attacks. We attribute this performance drop to the fact that the SVHN, MNIST, and Gaussian Noise distributions are more distant from CIFAR10 (the inlier distribution here) compared to Tiny ImageNet. Our results underscore the importance of using an auxiliary outlier dataset with a distribution closely related to the inlier distribution during adversarial training for robust OOD detection.

from a random and fixed dataset and are leveraged during training as samples of OOD. It is clear that these samples should be semantically different from the inlier training set to avoid misleading the detection.

We assert that in adversarial settings, the OE technique's performance is highly sensitive to the distance between the exposed outliers and the inlier training set distributions. As illustrated in Fig. 1, when two SOTA robust detection methods, ALOE (15) and ATD (13), utilize Tiny ImageNet (22) as auxiliary outlier samples for the CIFAR10 vs. CIFAR100 OOD detection task, the clean detection and more significantly adversarial detection performance improves compared to using MNIST (23) as the auxiliary outlier dataset. This suggests that an OE dataset closer to the in-distribution is significantly more beneficial than a distant one. This observation aligns with (24), which suggests that incorporating data near the decision boundary leads to a more adversarially robust model in the classification task.

Simultaneously, numerous studies (25; 26) have demonstrated that adversarial training demands a greater level of sample complexity relative to clean settings. Thus, several pioneering efforts have been made to enrich the data diversity, either by incorporating synthetic data (27; 28; 29; 30; 31; 32), or utilizing augmentation techniques (33; 34; 35; 36; 37; 38) to enhance the adversarial robustness. These observations prompt us to propose the following: *For adversarial training to be effective in robust OOD detection, the set of outliers in OE methods needs to be diverse," near-distribution," and "conceptually distinguishable" from the inlier samples.*

By 'near-distribution outliers,' we refer to data that possess semantically and stylistically related characteristics to those of the inlier dataset. If the outliers are not near-distribution, the classification boundary would be misplaced in the input space, causing a distribution shift on the adversarial perturbations of the outlier samples compared to those of the real anomalies. We have conducted numerous extensive ablation studies (sec. 5), and provided theoretical insights (sec. 2, 10) to offer evidence for these claims. Driven by the mentioned insights, we propose *Robust Out-of-Distribution Detection via Exposing adaptive Outliers* (RODEO), a method that leverages an adaptive OE strategy. This strategy is a data generation technique that conditions the generation process based on information from the in-distribution samples.

Although some previous works (13; 20; 39; 40) have proposed conditional generation techniques for crafting more adaptable auxiliary outliers compared to traditional methods using random outlier

datasets, these efforts have not yielded satisfactory results. Our hypothesis is that this can be attributed to their limited consideration of information solely in the image domain for conditioning. In contrast, we leveraged all available information in the in-distribution, including both images and text for this purpose. Our ablation studies (sec. 5) demonstrate that excluding any of this information leads to a decrease in performance.

We find that existing representative text-to-image models like Stable Diffusion (41) and DreamBooth (42) fall short of satisfying our criteria for synthesized OE (sec. 5), despite their high training complexity. This is likely because these models are optimized for general text-to-image generation rather than auxiliary outlier generation specifically. To address this limitation, we propose a novel OE generation pipeline specifically designed to synthesize effective outliers that improve robustness in OOD detection. In our proposed method, we employ a pre-trained generative diffusion model (43) and combine it with a CLIP (44) model to generate near-distribution outlier data. To be more precise, based on the inlier samples label(s) (e.g., "Dog"), we use a simple text encoder to extract words that are semantically similar to the inlier class labels. Then, we drop the ones that are not distinct from inlier labels using a threshold that is computed using the validation set. Next, using these near-OOD words (e.g. "Wolf"), the CLIP model guides the generative diffusion model to generate near-distribution OOD data, conditioned on the in-distribution images. Finally, we filter generated images that belong to the in-distribution with another threshold that is computed by the CLIP score and a validation set.

Our key idea here is that crafting auxiliary outliers is often made more flexible through text guidance. Through extensive experimentation, we demonstrate that our proposed generation scheme meets the criteria of diversity, proximity to the in-distribution, and distinctiveness from inlier samples, all concurrently. This, therefore, facilitates the effective use of adversarial training to achieve robust detection methods. We summarize the main contributions of this paper as follows:

- We have demonstrated the significance of considering both near and diverse auxiliary OOD samples to enhance robust outlier detection through theoretical insights and comprehensive experimental evidence. Furthermore, we have conducted an ablation study to show that our generated data outperforms alternative OE methods in terms of the mentioned characteristics.
- Our proposed method, RODEO, achieves significant results on various datasets, including medical and tiny datasets, highlighting the applicability of our work to real-world applications. Our method achieves competitive results in clean settings and establishes a SOTA performance in adversarial settings, surpassing existing methods by up to 50% in terms of AUROC.
- Interestingly, RODEO demonstrates generality in its application to various outlier detection setups, including ND, OSR, and OOD detection. Notably, previous works have primarily been limited to specific types of outlier detection setups.

## 1 BACKGROUND AND RELATED WORK

**Baselines** Several works have been proposed in outlier detection, with the goal of learning the distribution of normal samples, some methods such as DeepSVDD (45) and CSI (46) do this with self-supervised approaches. On the other hand many methods such as DN2 (47), PANDA (48), MSAD (49), Transformaly (11), ViT-MSP (50) and Patchcore (51) aim to leverage the knowledge from the pre-trained models. Furthermore, some other works have pursued outlier detection in an adversarial setting, including APAE (19), PrincipaLS (14), OCSDF (18), and OSAD (17). ATOM (20), ALOE (15), and ATD (13) achieved relatively better results compared to others by incorporating OE techniques and adversarial training. However, their performance falls short (as presented in Fig. 1) when the normal set distribution is far from their fixed OE set. For more details about previous works see Appendix (sec. 11).

**Adversarial Attacks** For the input $x$ with an associated ground-truth label $y$, an adversarial example $x^*$ is generated by adding a small noise to $x$, maximizing the predictor model loss $\ell(x^*; y)$. Projected Gradient Descent (PGD) (52) method is regarded as a standard and effective attack technique that functions by iteratively maximizing the loss function, through updating the perturbed input by a step size $\alpha$ in the direction of the gradient sign of $\ell(x^*; y)$ with respect to

$x$: $x_0^* = x$, $x_{t+1}^* = x_t^* + \alpha \cdot \text{sign}\left(\nabla_x \ell\left(x_t^*, y\right)\right)$, where the noise is projected onto the $\ell_\infty$-ball with a radius of $\epsilon$ during each step.

**Denoising Diffusion Probabilistic Models (DDPMs)** DDPMs (53; 54) are trained to reverse a parameterized Markovian process that transforms an image to pure noise gradually over time. Beginning with isotropic Gaussian noise samples, they iteratively denoise the image and finally convert it into an image from the training distribution. In particular a network employed and trained as follows: $p_\theta(x_{t-1}|x_t) = \mathcal{N}(\mu_\theta(x_t, t), \Sigma_\theta(x_t, t))$. This network takes the noisy image $x_t$ and the embedding at time step $t$ as input and learns to predict the mean $\mu_\theta(x_t, t)$ and the covariance $\Sigma_\theta(x_t, t)$. Recent studies have shown that DDPMs can be utilized for tasks such as generating high-quality images, as well as for editing and inpainting (43; 55; 56).

## 2 THEORETICAL INSIGHTS

In this section, we discuss the benefits of using near-distribution outliers over far-distribution outliers in the adversarial settings. For a better understanding, we consider a simplified example illustrated in Fig. 2.

Consider a one-dimensional feature space $\mathbb{R}$. The normal class follows $U(0, a - \epsilon)$, while the anomaly class adheres to $U(a + \epsilon, b)$. We aim to construct a robust anomaly detector that can handle $\ell_2$ perturbations of norm at most $\epsilon$ via OE. Here, we assumed the OE distribution to have a safe margin from the normal training set, to ensure that a small adversarial training loss is achievable. That is, we assume OE to follow $U(a + r, c)$, where $r \geq \epsilon$.

In this scenario, the optimal threshold $k$ for classification of the normal and the exposed outliers in the adversarial training scenario satisfies $a \leq k \leq a + r - \epsilon$, when a large sample size is available. Note that as the OE samples act as a proxy for the anomaly class, and hence could be shifted away to the right, the threshold tends to be placed to the right, i.e. close to anomalies rather than the normal samples. For equally weighted normal and anomaly classes, the adversarial test error rate would be:

$$\frac{\min(k + \epsilon, b) - a - \epsilon}{b - a - \epsilon} \tag{1}$$

Key observations include:

- For $k = a$, the adversarial error rate is zero.
- For $k > a$, the errors manifest in the interval $(a + \epsilon, \min(k + \epsilon, b))$.

Our analysis, complemented by Fig. 2, reveals that as the adversarial test error scales with $k$ when $k \neq a$, and hence with $r$, setting $r$ to its smallest value (i.e. $r \to \epsilon$) minimizes the adversarial error rate. Therefore, using near-distribution outliers in adversarial settings is not only advisable but also impactful. For a broader and more complete explanation, please refer to the Appendix (sec. 10).

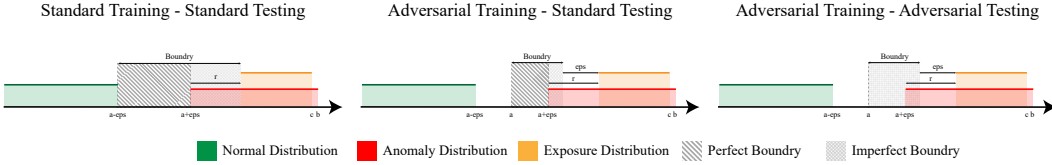

Figure 2: Effect of the outlier exposure distribution in a one-dimensional feature space in the adversarial and standard setups. Gray regions indicate feasible thresholds for separating the classes in the training data, where green represents the normal class and orange represents the outlier exposure. The parameter $r$ measures the shift in exposed outliers relative to the actual anomalies, and bold grays indicate thresholds yielding perfect test AUROC values. **Left:** Standard training scenario with many perfect thresholds, even with distant outlier exposure. **Middle:** Adversarial training narrows the set of perfect thresholds by reducing the feasible options. **Right:** During adversarial testing, the set of perfect thresholds contracts to a single point $a$, highlighting the criticality of near-distribution exposure in adversarial setups.

# 3 METHOD

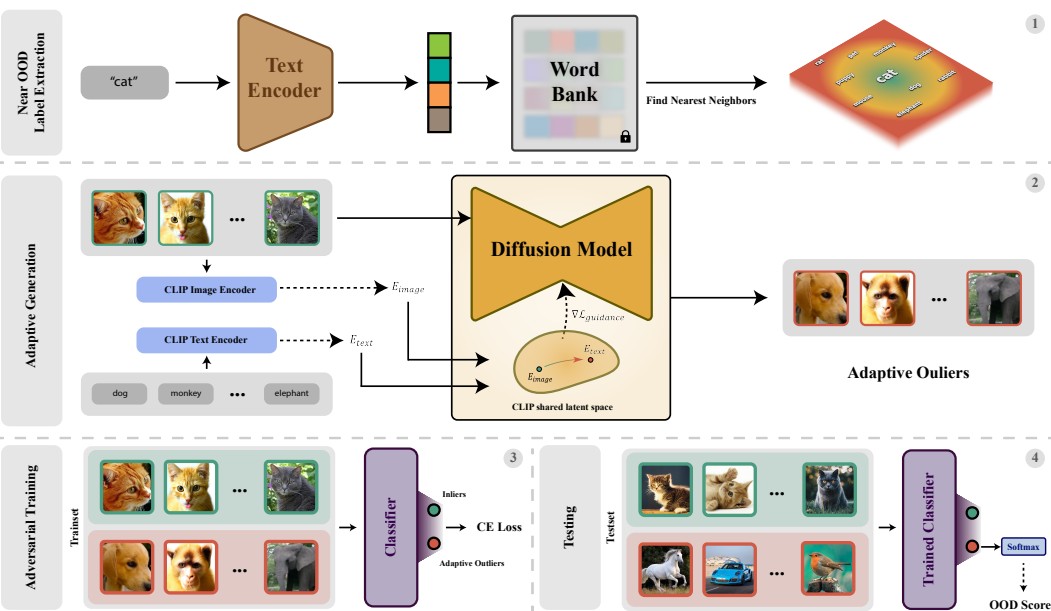

Figure 3: Our proposed adversarially robust outlier detection method is initiated with a Near Distribution Label Lookup, which finds words analogous to a given input label. These words, combined with inlier training image data, are employed in the Adaptive Generation stage to create near-OOD data. This stage is followed by adversarial training using both inlier and generated OE data, utilizing the cross-entropy loss function. During testing, the model processes the input and computes the OOD score as the softmax of the OOD class. (The data filtering steps are not shown in this figure)

**Motivation** To develop a robust outlier detection model, the OE technique appears to be crucial (20), otherwise, the model would lack information about the adversarial patterns on the outlier data, leaving it highly vulnerable to the adversarial attacks targeting the OOD data during test time. However, the vanilla OE technique, which involves leveraging outliers from a presumed dataset, leads to unsatisfactory results in situations where the outliers deviate significantly from a normal distribution (see Fig. 2). Motivated by these factors, we aim to propose an adaptive OE technique that attempts to generate diverse and near-distribution outliers, which can act as a proxy for the real inference-time outliers. The subsequent sections will provide a detailed description of the primary stages of our methodology: generation and filtering processes, training, and testing, each with their respective elements. Our method is outlined in Fig. 3.

## Generation Step

**Near-OOD (NOOD) Label Extraction** Employing a text extractor model and given the class labels of the inliers, we can identify words closely related to them. We utilize Word2Vec (57), a simple and popular model, to obtain the embeddings of the inlier labels denoted as $y_{\text{normal}}$ (e.g. *"screw"*) and subsequently retrieve their nearest labels (e.g. *"nail"*). In the next step, we employ ImageNet labels as the validation set and the CLIP text encoder to compute a threshold and refine the extracted labels by eliminating those exhibiting semantic equality to the inlier label. We then add some texts containing negative attributes of the inlier labels to the extracted set (e.g. *"broken screw"*). The union of these two sets of labels forms $y_{\text{NOOD}}$, which will guide the image generation process that utilizes the CLIP model. More details about NOOD label generation and threshold ($\tau_{Label}$) computing are available in Appendix (sec. 18).

**CLIP Guidance** The CLIP model is designed to learn joint representations between the text and image domains, and it comprises a pre-trained text encoder and an image encoder. The CLIP model operates by embedding both images and text into a shared latent space. This allows the model to assign a CLIP score, that evaluates the relevance of a caption to the actual content of an image. In order to extract knowledge effectively from the CLIP in image generation, we propose the

$\mathcal{L}_{\text{guidance}}\left(\boldsymbol{x}_{\text{gen}}, \boldsymbol{y}_{\text{NOOD}}\right)$ loss which aims to minimize the cosine distance between the CLIP space embeddings of the generated image $x_{\text{gen}}$ and the target text $\boldsymbol{y}_{\text{NOOD}}$:

$$\mathcal{D}\left(x, y\right) = \frac{x^{\top} y}{\|x\|\|y\|}, \quad \mathcal{L}_{\text{guidance}}\left(\boldsymbol{x}_{\text{gen}}, y_{\text{NOOD}}\right) = -\mathcal{D}\left(E_I\left(x_{\text{gen}}\right), E_T\left(y_{\text{NOOD}}\right)\right) \tag{2}$$

Here, $E_I$ and $E_T$ represent the embeddings extracted by the CLIP image and text encoders, respectively. During the conditional diffusion sampling process, the gradients from the $\mathcal{L}_{\text{guidance}}$ will be used to guide the normal samples towards the near outliers.

**Conditioning on Image** To generate near OOD data with shared semantic and stylistic features resembling a normal distribution, the generation process should incorporate the normal distribution. Consequently, we condition the denoising process on the normal images instead of initializing it with random Gaussian noise. More specifically, we employ a pre-trained diffusion generator and initiate the reverse process from a random time step $t_0 \sim U(0, T)$. More precisely, instead of beginning from $t_0 = T$, we start the diffusion process with the normal data augmented with noise. We then progressively remove the noise with the CLIP guidance to obtain a denoised result that is out-of-distribution (OOD) and close to a normal distribution: $x_{t-1} \sim \mathcal{N}(\mu(x_t|y_{\text{NOOD}}) + s \cdot \Sigma(x_t|y_{\text{NOOD}}) \cdot \nabla_{x_t}(\mathcal{L}_{\text{guidance}}(x_t, y_{\text{NOOD}})), \Sigma(x_t|y_{\text{NOOD}}))$, the scale coefficient $s$ controls the level of perturbation applied to the model. This type of conditioning leads to the generation of diverse outliers since, with smaller $t_{init}$, normal images would undergo small changes (pixel-level outliers), while relatively larger $t_0$ values lead to larger changes (semantic-level outliers). Please see Appendix (sec. 12.1) for more details about the generation step, and refer to Figures 9 for some generated images.

**Data Filtering** There is a concern that the generated images may still belong to the normal distribution, which can potentially result in misleading information in the subsequent steps. To mitigate this issue, we have implemented a method that involves defining a threshold to identify and exclude data that falls within the inlier distribution. To determine the threshold, we make use of the ImageNet dataset and CLIP score to quantify the mismatch. We calculate the CLIP score for the synthesized data and its corresponding normal label. If the computed CLIP score exceeds the threshold, it indicates that the generated data would likely belong to the normal distribution and should be excluded from the outlier set. Assuming the ImageNet dataset includes $M$ classes and each class includes $N$ data samples, let $\mathcal{X} = \{x_1^1, x_2^1, \ldots, x_N^M\}$ be the set of all data samples and $\mathcal{Y} = \{y_1, \ldots, y_M\}$ be the set of all labels, where $x_k^l$ indicates the $k^{\text{th}}$ data sample with label $y_l$. The threshold is then defined as:

$$\tau_{Image} = \frac{\sum_{i=1}^{N} \sum_{j=1}^{M} \sum_{r=1, r \neq j}^{M} \mathcal{D}\left(E_I(x_i^j), E_T(y_r)\right)}{MN(M-1)} \tag{3}$$

**Model Selection** During our OE generation process, CLIP encoders get input $x_t$ which is a noisy image and the public CLIP model is trained on noise-free images, this leads to generating low-quality data, as observed in (58). As a result, we utilize the small CLIP model proposed by (58), which has been trained on noisy image datasets. It is worth noting this model has been trained on 67 million samples, but is still well-suited for our pipeline and can generate OOD samples that it has not been trained on during training. See Appendix (sec. 15) for more details.

## Training Step

**Adversarial Training** During training, we have access to an inlier dataset $\mathcal{D}^{in}$ of pairs $(x_i, y_i)$ where $y_i \in \{1, ..., K\}$, and we augment it with generated OE $D^{gen}$ of pairs $(x_i, K+1)$ as outlier exposures to obtain $\mathcal{D}^{train} = \mathcal{D}^{in} \cup D^{gen}$. Then, we adversarially train a classifier $f_\theta$ with the standard cross-entropy loss $\ell_\theta$: $\min_\theta E_{(x,y) \sim \mathcal{D}_{train}} \max_{\|x^* - x\|_\infty \leq \epsilon} \ell_\theta(x^*, y)$, with the minimization and maximization done respectively by Adam and PGD-10. For evaluation purposes, we utilize a dataset $D^{test}$ that consists of both inlier and outlier samples.

## Test Step

**Adversarial Testing** During test time, we utilize the $(K+1)$-th logit of $f_\theta$ as the OOD score, which corresponds to the class of outliers in the training dataset. For the evaluation of our model, as well as other methods, we specifically target both in-distribution and OOD samples with several end-to-end adversarial attacks. Our objective is to cause the detectors to produce erroneous detection results by decreasing OOD scores for the outlier samples and increasing the OOD scores for the normal samples. We set the value of $\epsilon$ to $\frac{8}{255}$ for low-resolution datasets and $\frac{2}{255}$ for high-resolution ones. For the PGD attack, we use a single random restart for the attack, with random initialization within the range of $(-\epsilon, \epsilon)$, and perform $N = 100$ steps. Furthermore, we select the attack step size as $\alpha = 2.5 \times \frac{\epsilon}{N}$. In addition to the PGD attack, we have evaluated the models using AutoAttack (59), which is a union of a set of strong attacks designed to enhance the robustness evaluation. Furthermore, in the Appendix (see Table 8), we have evaluated the models under black-box attack (60). For additional information on the evaluation metrics, datasets, and implementation, please refer to the Appendix (sec. 7).

# 4 EXPERIMENTAL RESULTS

In this section, we conduct comprehensive experiments to evaluate previous outlier detection methods, including both standard and adversarially trained approaches, as well as our own method, under clean and various adversarial attack scenarios. Our experiments are organized into three categories and are presented in Tables 1, 2, and 3. We provide a brief overview of each category below, and further details can be found in the Appendix (sec. 19, 20).

**Novelty Detection** For each dataset containing $N$ distinct classes, we perform $N$ separate experiments and average the results. In every individual experiment, samples from a single class are used as the normal data, while instances from the remaining $N - 1$ classes are considered outliers. Table 1 presents the results of this setup.

**Open-Set Recognition** For this task, each dataset was randomly split into a normal set and an outlier set at a 60/40 ratio. This random splitting was repeated 5 times. The model was exclusively trained on the normal set samples, with average AUROC scores reported. Results are presented in Table 3.

**Out-Of-Distribution Detection** For the task of OOD detection, we considered CIFAR10 and CIFAR100 datasets as normal datasets in separate experiments. Following earlier works (15), and (21), we test the model against several out-of-distribution datasets which are semantically distinct from the in-distribution datasets, including MNIST, TinyImageNet (22), Places365 (61), LSUN (62), iSUN (63), Birds (64), Flowers (65), COIL-100 (66) and CIFAR10/CIFAR100 (depending on which is considered the normal dataset). For any given in-distribution dataset, the results are averaged over the OOD datasets. We have provided the results of this task in Table 2.

Table 1: AUROC (%) performance of ND methods on various datasets in a clean setting and under adversarial attacks using PGD100 and AutoAttack (AA). Perturbations for attacks are $\epsilon = \frac{8}{255}$ for low-resolution datasets and $\epsilon = \frac{2}{255}$ for high-resolution datasets. The best results are **bolded** and the second-best results are underlined.

| | Dataset | Attack | DeepSVDD | CSI | MSAD | Transformaly | PatchCore | PrincipaLS | OCSDF | APAE | RODEO (Ours) |
|---|---|---|---|---|---|---|---|---|---|---|---|
| **Low-Res** | CIFAR10 | Clean | 64.8 | 94.3 | 97.2 | **98.3** | 68.3 | 57.7 | 57.1 | 55.2 | 87.4 |
| | | PGD / AA | 23.4 / 9.7 | 2.8 / 0.0 | 0.0 / 0.0 | 0.0 / 0.0 | 0.3 / 0.0 | 24.1 / 20.2 | 22.0 / 15.3 | 0.1 / 0.0 | **71.1 / 69.3** |
| | CIFAR100 | Clean | 67.0 | 89.6 | 96.4 | **97.3** | 66.8 | 52.0 | 48.2 | 51.8 | 79.6 |
| | | PGD / AA | 14.5 / 5.8 | 3.2 / 0.0 | 2.9 / 4.3 | 4.3 / 2.6 | 0.0 / 0.0 | 16.6 / 14.7 | 15.1 / 12.0 | 0.0 / 0.0 | **62.8 / 61.0** |
| | MNIST | Clean | 94.8 | 93.8 | 96.0 | 94.8 | 83.2 | 97.3 | 95.5 | 92.5 | **99.4** |
| | | PGD / AA | 11.4 / 9.6 | 0.3 / 0.4 | 10.8 / 7.3 | 10.8 / 6.7 | 0.0 / 0.0 | 78.1 / 72.5 | 62.4 / 58.3 | 22.9 / 19.8 | **95.7 / 95.2** |
| | Fashion-MNIST | Clean | 94.5 | 92.7 | 94.2 | 94.4 | 77.4 | 91.0 | 90.6 | 86.1 | **95.6** |
| | | PGD / AA | 49.4 / 38.2 | 4.8 / 3.1 | 0.0 / 0.0 | 0.5 / 0.0 | 0.0 / 0.0 | 61.3 / 58.2 | 54.5 / 49.2 | 11.2 / 7.0 | **88.1 / 87.6** |
| | SVHN | Clean | 60.3 | 96.0 | 63.1 | 55.4 | 52.1 | 63.0 | 58.1 | 52.6 | **78.6** |
| | | PGD / AA | 7.7 / 2.8 | 1.3 / 0.4 | 0.6 / 2.5 | 7.6 / 5.8 | 3.2 / 0.1 | 31.4 / 25.0 | 24.9 / 23.5 | 17.4 / 19.8 | **45.4 / 41.2** |
| **High-Res** | MVTecAD | Clean | 67.0 | 63.6 | 87.2 | 87.9 | **99.6** | 63.8 | 58.7 | 62.1 | 61.5 |
| | | PGD / AA | 2.6 / 0.0 | 0.0 / 0.0 | 0.9 / 0.0 | 0.0 / 0.0 | 7.2 / 4.8 | **24.3** / 12.6 | 5.2 / 0.3 | 4.7 / 1.8 | 15.9 / **14.2** |
| | Head-CT | Clean | 62.5 | 60.9 | 59.4 | 78.1 | **98.5** | 68.9 | 62.4 | 68.1 | 87.3 |
| | | PGD / AA | 0.0 / 0.0 | 0.6 / 0.0 | 0.0 / 0.0 | 6.4 / 3.2 | 1.5 / 0.0 | 27.8 / 16.2 | 13.1 / 8.5 | 6.6 / 3.8 | **70.0 / 68.4** |
| | BrainMRI | Clean | 74.5 | 93.2 | **99.9** | 98.3 | 91.4 | 70.2 | 63.2 | 55.4 | 76.3 |
| | | PGD / AA | 4.3 / 2.1 | 0.0 / 0.0 | 1.7 / 0.0 | 5.2 / 1.6 | 0.0 / 0.4 | 33.5 / 17.8 | 20.4 / 12.5 | 9.7 / 8.3 | **71.1 / 70.5** |
| | Tumor Detection | Clean | 70.8 | 85.3 | 95.1 | **97.4** | 92.8 | 73.5 | 65.2 | 64.6 | 89.0 |
| | | PGD / AA | 1.7 / 0.0 | 0.0 / 2.2 | 0.1 / 0.0 | 7.4 / 5.1 | 9.3 / 6.1 | 25.2 / 14.7 | 17.9 / 10.1 | 15.8 / 8.3 | **67.5 / 66.9** |
| | Covid19 | Clean | 61.9 | 65.1 | 89.2 | **91.0** | 77.7 | 54.2 | 46.1 | 50.7 | 79.6 |
| | | PGD / AA | 0.0 / 0.0 | 0.0 / 0.3 | 4.7 / 1.9 | 10.6 / 4.4 | 4.2 / 0.5 | 15.3 / 9.1 | 9.0 / 6.5 | 11.2 / 8.7 | **59.4 / 58.8** |
| | **Mean** | Clean | 71.8 | 83.4 | 87.8 | **89.3** | 80.8 | 69.2 | 64.5 | 63.9 | 83.4 |
| | | PGD / AA | 11.5 / 6.8 | 1.3 / 0.6 | 1.1 / 1.6 | 5.3 / 2.9 | 2.6 / 1.2 | 33.8 / 26.1 | 24.4 / 19.6 | 10.0 / 7.7 | **64.7 / 63.3** |

Table 2: AUROC (%) performance of OOD detection methods, where CIFAR-10 and CIFAR-100 are considered as separate inlier datasets in each experiment. The union of other datasets are used as OOD data. This evaluation considers both the PGD-100 attacked and clean setups. Perturbations for attacks are $\epsilon = \frac{8}{255}$.

| | | Method | | | | | |
|---|---|---|---|---|---|---|---|
| In-Dataset | Attack | ViT-MSP | AT* | ATOM | ALOE | ATD | RODEO |
| CIFAR10 | Clean | **99.5** | 80.5 | 82.7 | 97.8 | 94.3 | 93.2 |
| | PGD | 0.0 | 20.8 | 25.1 | 6.0 | 69.3 | 70.4 |
| CIFAR100 | Clean | **95.1** | 70.0 | 91.6 | 79.3 | 87.7 | 88.1 |
| | PGD | 0.0 | 13.6 | 5.4 | 26.4 | 55.3 | 66.4 |

* AT indicates the model was trained without using OE.

Table 3: AUROC (%) performance of OSR methods on various datasets in a clean setting and under PGD-100 attack.

| | | Method | | | | | |
|---|---|---|---|---|---|---|---|
| Dataset | Attack | ViT-MSP | AT* | ATOM | ALOE | ATD | RODEO |
| MNIST | Clean | 92.4 | 80.2 | 74.8 | 79.5 | 68.7 | **97.2** |
| | PGD | 4.2 | 36.2 | 6.3 | 38.2 | 56.7 | **88.8** |
| FMNIST | Clean | 87.6 | 72.5 | 64.3 | 72.6 | 59.6 | **87.7** |
| | PGD | 3.1 | 31.7 | 4.7 | 29.0 | 43.0 | **67.1** |
| CIFAR10 | Clean | **96.8** | 65.2 | 68.3 | 52.4 | 49.0 | 79.6 |
| | PGD | 2.5 | 21.0 | 5.2 | 25.7 | 33.6 | **48.5** |
| CIFAR100 | Clean | **92.1** | 61.7 | 51.4 | 49.8 | 50.5 | 64.1 |
| | PGD | 0.0 | 18.3 | 3.2 | 18.6 | 36.6 | **37.7** |

* AT indicates the model was trained without using OE.

As the results indicate, our method demonstrates significant performance in adversarially robust outlier detection, outperforming others by a large margin under various strong attacks. Notably, in open-world applications where robustness is crucial, a slight decrease in clean performance is an acceptable trade-off for enhanced robustness. Our results support this stance, achieving an average of 83.4% in clean settings and 64.7% in adversarial scenarios across various datasets. This performance surpasses SOTA methods in clean detection like Transformaly, which, while achieving 89.3% in clean settings, experiences a substantial drop to 5.3% in adversarial conditions.

Table 4: AUROC (%) of the detector model after adversarial training with outliers generated by different OE techniques in clean data and under PGD-100 attack evaluation. The results indicate that our adaptive OE method outperforms other methods in terms of improving robust detection. The experiments were conducted in the ND setting (clean/PGD-100).

| Exposure Technique | Target Dataset | | | | | | Mean |
|---|---|---|---|---|---|---|---|
| | CIFAR10 | MNIST | FMNIST | MVTec-ad | Head-CT | Covid19 | |
| Gaussian Noise | 54.4 / 11.3 | 56.1 / 12.4 | 52.7 / 15.7 | 47.9 / 0.1 | 49.0 / 0.8 | 50.7 / 0.0 | 51.8 / 7.4 |
| Vanilla OE (ImageNet) | 87.3 / 70.0 | 90.0 / 43.0 | 93.0 / 82.0 | **64.6** / 0.5 | 61.8 / 2.1 | 62.7 / 24.5 | 75.6 / 34.6 |
| Mixup with ImageNet | 59.4 / 31.5 | 59.6 / 1.7 | 74.2 / 48.8 | 58.5 / 1.4 | 54.4 21.4 | 69.2 / 50.8 | 62.8 / 27.6 |
| Fake Image Generation | 29.5 / 16.2 | 76.0 / 51.3 | 52.2 / 31.1 | 43.5 / 7.3 | 63.7 / 6.9 | 42.7 / 13.0 | 51.2 / 22.5 |
| Stable Diffusion Prompt | 62.4 / 35.6 | 84.3 / 62.5 | 63.7 / 48.5 | 54.9 / 12.6 | 71.5 / 3.6 | 37.1 / 0.0 | 60.1 / 23.5 |
| **Adaptive OE (Ours)** | **87.4 / 71.1** | **99.4 / 95.7** | **95.6 / 88.1** | 61.5 / **15.9** | **87.3 / 70.0** | **79.6 / 59.4** | **84.0 / 66.8** |

Table 5: Comparison of generated outliers by different OE techniques using the $FDC$ metric. A higher value indicates that the generated outliers have more diversity and are closer to the corresponding normal dataset (target dataset). The experiments were conducted in the ND setting.

| Exposure Technique | Target Dataset | | | | | | Mean |
|---|---|---|---|---|---|---|---|
| | CIFAR10 | MNIST | FMNIST | MVTec-ad | Head-CT | Covid19 | |
| Gaussian Noise | 0.822 | 0.704 | 0.810 | 0.793 | 0.735 | 0.674 | 0.756 |
| Vanilla OE (ImageNet) | 7.647 | 0.795 | 2.444 | 1.858 | 3.120 | 3.906 | 3.295 |
| Mixup with ImageNet | 2.185 | 0.577 | 0.985 | 1.866 | 1.587 | 2.078 | 1.547 |
| Fake Image Generation | 0.987 | 0.649 | 0.907 | 1.562 | 0.881 | 0.896 | 0.980 |
| Stable Diffusion Prompt | 1.359 | 1.790 | 1.399 | **1.982** | 1.097 | 0.654 | 1.381 |
| **Adaptive OE (Ours)** | **8.504** | **11.395** | **3.819** | 1.948 | **12.016** | **5.965** | **6.687** |

# 5 ABLATION STUDIES

In this section, we present two quantitative approaches to compare our adaptive OE method with alternative techniques. Also, to provide more intuition, we provided a qualitative evaluation presented in Fig. 4. Firstly, to demonstrate the superiority of our adaptive OE method in terms of improving robust detection, we conducted an experiment where we replaced the outliers generated by our method with outliers generated using other techniques. We then trained the detector model from scratch and re-evaluated the model. The results of this comparison are presented in Table 4, clearly indicating the significant effectiveness of our method.

Secondly, we introduce another quantitative approach that highlights the superiority of our generated outlier data in terms of diversity and its proximity to the normal sample distribution using common benchmarks in the image generation field. Specifically, we employed the Fréchet Inception Distance (FID) metric (67) to measure the distance between the generated outliers and the normal distribution. Additionally, we used the Density and Coverage metrics, proposed by (68), to evaluate the diversity of the generated data. For more details on the FID, Density, and Coverage metrics used in our analysis, refer to Appendix (sec. 16). For a unified comparison of outlier exposure methods in terms of proximity and diversity, we defined $FDC \propto \frac{\log(Density) + \log(Coverage)}{\log(FID)}$ for each generated outlier set and its corresponding normal set. Higher FID values indicate a greater distance to the normal distribution, and a higher Density & Convergence metric indicates more diversity. As a result, a higher FDC value indicates that the generated outliers have more diversity and are closer to the normal distribution. The results of this experiment are presented in Table 5, indicating our adaptive OE method results in near and diverse OOD. More detailed results of this experiment are in Appendix (see Table 9). In the following, we will briefly explain alternative OE methods that we considered in our ablation study.

**Alternative OE Methods** Vanilla OE refers to experiments where we utilize a fixed dataset, e.g. ImageNet, to provide the OE. 'Mixup with ImageNet' is referred to as a more adaptive OE technique, wherein instead of

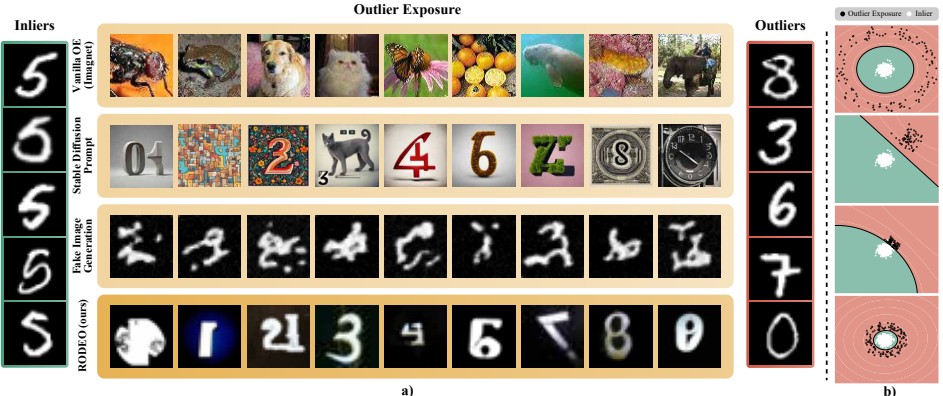

Figure 4: This figure provides an overview of outlier data from different OE techniques. The second and third rows present methods that consider either text or image domain information exclusively. In contrast, our proposed method, RODEO, shown in the last row, incorporates information from both domains simultaneously. RODEO demonstrates the ability to alter semantic content while maintaining visual similarity between the original and edited images. To provide further intuition about the importance of diversity and the distance of OE from the normal distribution, we compute features for normal and generated outlier data via a pretrained ViT model (72), and apply t-SNE (73) to reduce the features to 2D. We then find decision boundaries of the data with SVM (74) and present them on the right side of each generated OE example.

using ImageNet data as OE, it combines ImageNet samples with normal samples through mixup to generate OE samples that are closer to the normal distribution, as proposed in (69). Fake Image Generation' proposed in (39) refers to a technique for generating outliers that conditions a generative diffusion model on images without any text conditioning. This method utilizes a diffusion generator trained on the normal training set, but early stopped in the training, to generate synthetic images that resemble normal samples but have obvious distinctions from them. These are then treated as OE. In the 'Stable Diffusion Prompt' scenario, we utilize our near-OOD extracted text prompts and use them as prompts for the Stable Diffusion model to generate OE.

**Limitaion of SOTA Text-to-Image Diffusion Models** In our pipeline, conditioning the generator on images enables the synthesis of diverse OE data, including both pixel- and semantic-level OE. In contrast, SOTA text-to-image diffusion models, such as Stable Diffusion, operate on latent embeddings to reduce inference complexity. This makes them inappropriate for our goal, as it prevents us from generating pixel-level OE. Moreover, despite billions of training samples, these models exhibit biases (70) inherited from their training data (i.e. LAION (71)), which are ill-suited for our pipeline because our task involves applying it to datasets far from their training distribution, like medical imaging. On the other hand, our pipeline with just 67 million training samples is applicable to various datasets owing to its specific design. Qualitative images and more details can be found in the Appendix (sec. 15).

# 6 CONCLUSION

The combination of OE and adversarial training shows promising results in the robust OOD detection, however encounters significant performance drops in the case that OE dataset is far from the normal distribution. In light of these limitations, we introduce RODEO, a novel approach that addresses this challenge by focusing on improving the quality of outlier exposures. RODEO establishes a pipeline for generating diverse and near-distribution OOD synthetic samples through leveraging text descriptions of the potential OOD concepts and guiding diffusion model using these texts. These synthetic samples are then utilized as OE during the adversarial training of a discriminator. Through extensive evaluation, our approach outperforms existing methods, particularly excelling in the highly challenging task of Robust Novelty Detection. Notably, our approach maintains high OOD detection performance in both standard and adversarial settings across various detection scenarios, including medical and industrial datasets, demonstrating its high applicability in real-world contexts.

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

# Appendix

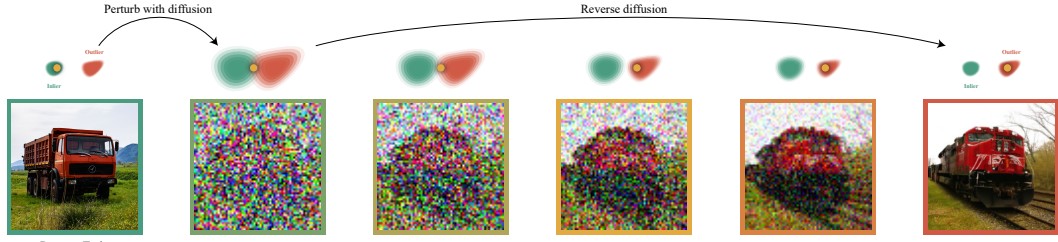

Prompt: Train

Figure 5: The figure illustrates a text-guided diffusion process. A yellow dot, representing an inlier data point within the green inlier distribution, is progressively transformed towards the red outlier distribution, driven by CLIP guidance. This showcases the model's ability to guide the transformation from inlier to outlier data via textual instructions.

## 7 EVALUATION METRICS & DATASETS & IMPLEMENTATION DETAILS

**Evaluation Metrics** AUROC is used as a well-known classification criterion. The AUROC value is in the range [0, 1], and the closer it is to 1, the better the classifier performance.

**Datasets** For the low-resolution datasets, we included CIFAR10(75), CIFAR100(75), MNIST(23), and FashionMNIST(76). Furthermore, we performed experiments on medical and industrial high-resolution datasets, namely Head-CT(77), MVTec-ad(78), Brain-MRI(79), Covid19(80), and Tumor Detection(81). The results are available in Table 1.

**Implementation Details** We use ResNet-18(82) as the architecture of our neural network for the high-resolution datasets and for the low-resolution datasets, we used Wide ResNet(83). Furthermore, RODEO is trained 100 epochs with Adam(84) optimizer with a learning rate of 0.001 for each experiment.

## 8 ALGORITHM

This algorithm presents the complete approach, including all components that are integral to it.

---

**Algorithm 1:** RODEO: Adv. Training with Adaptive Exposure Dataset

---

**Input:** $\mathcal{D}_{\text{in}}, \mathcal{D}_{\text{val}}, enc_{text}, \mu_{Diffusion}, \Sigma_{Diffusion}, E_I^{clip}, E_T^{clip}, f_\theta, K, T_0, T$ ; // $T_0 \in [0.3T, 0.6T]$
**Output:** $\hat{f}_\theta$
```
// Near-Distribution OOD Prompt Search
```
$\tau_{\text{label-val}} = Avg(Dist(E_T^{clip}(y_i), E_T^{clip}(y_j)))$ $\qquad\qquad\qquad\qquad \forall(y_i, y_j) \in \mathcal{Y}(\mathcal{D}_{\text{val}})$
**for** $(i, label) \in Y$ **do**
    $Prompts[i] \leftarrow enc_{text}.KNN(label)$
    $Prompts[i] \leftarrow Prompts[i].Remove(enc_{text}.MinDist(Prompt, Y \setminus label) < \tau_{\text{label-val}})$
    $Prompts[i] \leftarrow Prompts[i] \cup Append(NegativeAdjectives[label], label)$
**end**
```
// Adaptive Exposure Generation
```
$\tau_{\text{image-val}} = Avg(Dist(E_I^{clip}(x_i), E_T^{clip}(y)))$ $\qquad\qquad\qquad\qquad \forall(x_i, y_i) \in \mathcal{D}_{\text{val}} \forall y \neq y_i$
**for** $(x_i, y_i) \in \mathcal{D}_{in}$ **do**
    $c \sim \mathcal{U}(Prompts[y_i])$
    $t_{init} \sim \mathcal{U}([T_0, ..., T]$
    $\hat{x}_{t_{init}} = x_i$
    **for** $t = t_{init}, ..., 0$ **do**
        $\hat{\mu}(\hat{x}_t|c) = \mu_{Diffusion}(\hat{x}_t|c) + s \cdot \Sigma_{Diffusion}(\hat{x}_t|c) \cdot \nabla_{\hat{x}_t}(E_I^{clip}(\hat{x}_t) \cdot E_T^{clip}(c))$
        $\hat{x}_{t-1} \sim \mathcal{N}(\hat{\mu}(\hat{x}_t|c), \Sigma_{Diffusion}(\hat{x}_t|c))$
    **end**
```
        // Discarding too Similar Samples
```
    **if** $Dist(E_I^{clip}(\hat{x}_0), E_T^{clip}(y_i)) < \tau_{\text{image-val}}$ **then**
        $\mathcal{D}_{\text{exposure}} \leftarrow \mathcal{D}_{\text{exposure}} \cup \{(\hat{x}_0, K+1)\}$
    **end**
**end**
$\mathcal{D}_{\text{train}} \leftarrow \mathcal{D}_{\text{in}} \cup \mathcal{D}_{\text{exposure}}$
$\hat{f}_\theta \leftarrow \text{Adversarial-Training}(f_\theta, \mathcal{D}_{\text{train}})$

---

# 9 ND, OSR AND OOD DETECTION

As we have reported the results of our method on the most common settings for OOD detection, in this section, we provide a brief explanation for each setting to provide further clarification. In OSR, a model is trained on $K$ classes from an $N$-class training dataset. During testing, the model encounters $N$ distinct classes, where $N - K$ of these classes were not present during the training phase. ND is a type of open-set recognition that is considered an extreme case, specifically when $k$ is equal to 1. Some works refer to ND as one-class classification. OOD detection shares similarities with OSR; however, the key distinction is that the open-set and closed-set classes originate from two separate datasets. (6; 39)

# 10 A MORE THOROUGH THEORETICAL EXPLANATION

Let's assume that the inlier data is coming from $N(0, I)$ and the anomaly is distributed according to $N(a, I)$. Furthermore, let $N(a', I)$ be the outlier exposure data. We assume that the OE is farther away from the inlier class than the anomaly data, i.e. $\|a'\| \geq \|a\|$. Assuming access to large training set of inlier and exposure samples, the optimal classifier would be $y = \frac{a'^\top}{\|a'\|}(x - \frac{a'}{2}) = \frac{a'^\top}{\|a'\|}x - \frac{\|a'\|}{2}$, for an adversary that has a budget of at most $\epsilon$ perturbation in $\ell_2$ norm (25). Now, applying this classifier on the inlier and anomaly classes at the test time, we get:

$$\frac{a'^\top x}{\|a'\|} \sim N(0, I), \tag{4}$$

for a inlier $x$, and also:

$$\frac{a'^\top x}{\|a'\|} \sim N(\frac{a^\top a'}{\|a'\|}, I), \tag{5}$$

for an anomalous $x$. Therefore, using the trained classifier $y$ to discriminate the inlier and anomaly classes, the error rate would be:

$$(1 - \Phi(\frac{\|a'\|}{2} - \epsilon)) + (1 - \Phi(\frac{a^\top a'}{\|a'\|} - \frac{\|a'\|}{2} - \epsilon)), \tag{6}$$

where $\Phi(.)$ is the CDF for the inlier distribution $N(0, 1)$.

Let $\delta = a' - a$, and note that:

$$\frac{a^\top a'}{\|a'\|} - \frac{\|a'\|}{2} = \frac{(a' - \delta)^\top a'}{\|a'\|} - \frac{\|a'\|}{2} \tag{7}$$

$$= \frac{\|a'\|}{2} - \frac{\delta^\top a'}{\|a'\|}. \tag{8}$$

But note that:

$$\delta^\top a' = a'^\top a' - a^\top a' \tag{9}$$

$$= \|a'\|^2 - \|a\|\|a'\|\cos(\theta) \tag{10}$$

$$= \|a'\|(\|a'\| - \|a\|\cos(\theta)) \tag{11}$$

$$\geq \|a'\|(\|a'\| - \|a\|) \geq 0, \tag{12}$$

because we have previously assumed $\|a'\| \geq \|a\|$ to reflect that the OE could be far-distribution. Hence, note that the error rate can be written as:

$$\left(1 - \Phi\left(\frac{\|a'\|}{2} - \epsilon\right)\right) + \left(1 - \Phi\left(\frac{\|a'\|}{2} - \frac{\delta^\top a'}{\|a'\|} - \epsilon\right)\right) \tag{13}$$

$$= \left(1 - \Phi\left(\frac{\|a'\|}{2} - \epsilon\right)\right) + \left(1 - \Phi\left(\frac{\|a'\|}{2} - c - \epsilon\right)\right), \tag{14}$$

where $c \geq 0$. Note that for a fixed $\|a'\|$, by making $\cos(\theta)$ small, the error increases, as $\Phi$ is an increasing function. Also, note that for the case that $\theta = 0$, i.e. smallest possible error among fixed $\|a'\|$, the error can be rewritten as:

$$\left(1 - \Phi\left(\frac{\|a\|}{2} + \frac{(\|a'\| - \|a\|)}{2} - \epsilon\right)\right) + \left(1 - \Phi\left(\frac{\|a\|}{2} - \frac{(\|a'\| - \|a\|)}{2} - \epsilon\right)\right) \tag{15}$$

$$= \left(1 - \Phi\left(\frac{\|a\|}{2} + d - \epsilon\right)\right) + \left(1 - \Phi\left(\frac{\|a\|}{2} - d - \epsilon\right)\right), \tag{16}$$

with $d = \frac{(\|a'\| - \|a\|)}{2} \geq 0$. Note that if $d$ is close to zero, i.e. near-distribution OE, the error converges to that of the adversarial Bayes optimal. But as $d$ grows large, the error becomes larger. Therefore, the more OE is away from the inlier distribution, the larger the error rate becomes.

Now, let's assume that the OE follows a less *diverse* distribution, i.e. $N(a', \sigma^2 I)$, with $\sigma < 1$. In this case, the intercept of the optimal line that separates the two class gets biased towards the OE distribution, increasing the error rate of classifying normal vs. anomaly. Again, to make this error small, one has to increase $\sigma^2$ to a limit that matches the original anomaly distribution $\sigma^2 = 1$.

## 11   DETAILED BASELINES

Some works introduced the OE technique for OOD detection tasks, which utilizes auxiliary random images known to be anomalous (21). Many top-performing OOD detection methods incorporate OE to enhance their performance in both classic and adversarial OOD detection evaluation tasks (40; 10; 39). The most direct approach to utilizing outliers involves incorporating them into the training set, with labels uniformly selected from the label space of typical samples. In an effort to improve the adversarial robustness of OOD detection, some methods have attempted to make OE more adaptive. For example, ATD (13) employs a generator to craft fake features instead of images. Another approach, ALOE (20), mines low anomaly score data from an auxiliary OOD dataset for training, thereby enhancing the robustness of OOD detection.

Furthermore, some other works have pursued outlier detection in an adversarial setting which includes APAE (19), PrincipaLS (14) and OCSDF (18) and OSAD (17) ATOM(20) ALOE (15) and ATD (13), between these robust outlier detection methods, ATOM, ALOE and ATD achieved relatively better results by incorporating OE and adversarial training, however, their performance falls short in case that normal set distribution is far from their fixed OE set. For instance, APAE (19) suggested enhancing adversarial robustness through the utilization of approximate projection and feature weighting. PrincipaLS (14) proposed Principal Latent Space as a defense strategy to perform adversarially robust ND. OCSDF (18) aimed to achieve robustness in One-Class Classification (OCC) by learning a signed distance function to the boundary of the support of the normal distribution, which can be interpreted as the normality score. Through making the distance function $\ell_1$ Lipschitz, one could guarantee robustness against $\ell_2$ bounded perturbations. OSAD (17) augmented the model architecture with dual-attentive denoising layers, and integrated the adversarial training loss with an auto-encoder loss. The auto-encoder loss was designed to reconstruct the original image from its adversarial counterpart.

In the context of adversarial OOD scenarios, certain studies focused on utilizing the insights gained from the pre-trained models based on Vision Transformers (ViT) (50; 72). Some other works incorporated OE to enhance their performance in both clean and adversarial OOD detection evaluation tasks (20; 40; 10; 39). The most direct approach to utilizing outliers involved incorporating them into the training set, with labels uniformly selected from the label space of typical samples. In an effort to improve the adversarial robustness of the detection models, some methods have attempted to make OE more adaptive. For example, ATD (13) employed a generator to craft fake features instead of images, and applied adversarial training on OE and normal real samples to make the discriminator robust. Another approach, ATOM (20), mined low anomaly score data from an auxiliary OOD dataset for training, thereby enhancing the robustness of OOD detection through adversarial training on the mined samples.

## 12 DETAILS ABOUT EVALUATION AND GENERATION

In this section, we will provide more details about our evaluation methodology and Generation Step.

### 12.1 GENERATION STEP

In our proposed generation method, we perturb the in-distribution(ID) images with Gaussian noise and utilize a diffusion model with guidance from the extracted candidate near-OOD labels to shift the ID data to OOD data. This is possible because it has been shown that the reverse process can be solved not only from $t_0 = 1$ but also from any intermediate time (0, 1). We randomly choose an initial step for each data between 0 and 0.6, which is a common choice based on previous related works. (85; 86)

If we have k classes in the normal dataset, with each class containing N samples, we generate N OOD samples to extend the dataset to k+1 classes. However, if N is a small number (e.g. N<100), we may generate up to 3000 OOD samples to prevent overfitting.

### 12.2 ADVERSARIAL ATTACK ON OOD DETECTORS

Suppose we have a test dataset containing a data sample $x$ that belongs to either the OOD class (-1) or the ID class (1). We can also assume the existence of a trained OOD detector $O_{\theta} : \mathbb{R}^d \to \mathbb{R}$, which can evaluate an OOD score for each data sample. Depending on the label y, we can generate adversarial examples using the $l_{\infty}$-norm. This involves perturbing the input $l_{\infty}$ in such a way that modifies its OOD score either upwards or downwards:

$$x_0^* = x, \qquad x^{t+1} = x^t + \alpha \cdot \text{sgn}(\nabla_x (y.O_{\theta}(x^t))) \tag{17}$$

where $y = 1$ for in-distribution samples and $y = -1$ for OOD samples. We performed various strong attacks including PGD-1000 with 10 random restarts, Auto Attack and Adaptive Auto Attack (87). The latter is a recently introduced attack that has demonstrated considerable strength. The detailed experiments on these attacks is reported in Tables 11-13. It is also noteworthy that for Auto Attack, it was not possible to adapt the DLR(59) loss based attacks due to their presumption that the output of the model has at least 3 elements, which does not hold in OOD detection tasks.

### 12.3 COMPUTATIONAL COST

Experiments were conducted on RTX 3090 GPUs. Generating approximately 10,000 low-resolution and 1,000 high-resolution OOD (out-of-distribution) data required 1 hour. For the one-class anomaly detection, training each class of low-resolution datasets took about 100 minutes (see Figure 6 for detailed analysis). The OOD detection task required around 16 hours of training, and each experiment in the OSR (open-set-recognition) setting took approximately 9 hours.

## 13 USING IMAGE LABELS AS DESCRIPTORS

Novelty detection, also known as one-class classification, involves identifying instances that do not fit the pattern of a given class in a training set. Traditionally, methods for this task have been proposed without using the labels of the training data. For example, they did not take into account the fact that the normal set includes the semantic "dog". In the case of OOD detection (which is a multi-class setting), methods commonly extract features and define supervised algorithms using the labels of the normal set. However, they do not fully utilize the semantic information contained in these labels. Specifically, they only consider class labels as indexes for their defined task, such as classification.

Recently, there has been a growing interest in leveraging pre-trained multimodal models to enhance OOD detection performance, both in one-class and multi-class scenarios. Unlike prior works, these approaches utilize

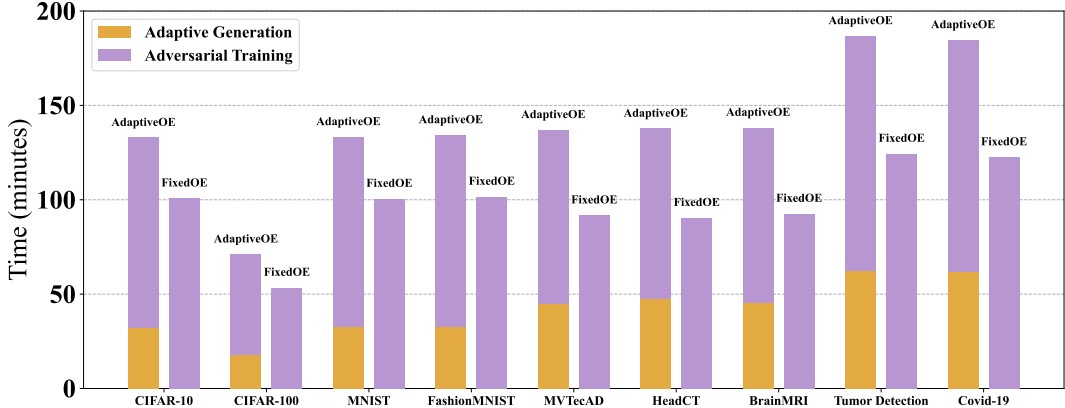

Figure 6: Comparative analysis of computational time for data generation and adversarial training across various datasets in one-class anomaly detection setting. The time is measured in minutes and is split into two components: data generation (golden segment) and the subsequent adversarial training phase (purple segment). The datasets range from standard image benchmarks like CIFAR-10 and MNIST to specialized medical and anomaly detection datasets such as MVTecAD, BrainMRI, and Covid-19.

the semantic information embedded within the labels of the normal set. This is akin to treating labels as image descriptors rather than just as indices. For example, (10) used CLIP in the novelty detection setting and utilized both pairs of normal images and their labels (e.g., a photo of x) to extract maximum information from the normal set. Similarly, (88) applied CLIP for zero-shot OOD detection and used both the image and semantic content of their respective labels to achieve the same goal. Motivated by these works, our study utilizes image labels as descriptors in all reported settings (ND, OSR, OOD). In fact, we utilized a simple language model to predict candidate unseen labels for OOD classes located near the boundary, leveraging these image labels.

**Discussion**    Although some recent works have used labels as descriptors, there may be concerns that this approach could provide unfair guidance since it is not commonly used in traditional literature. However, it is important to note that the OOD detection problem is a line of research with many practical applications in industries such as medicine autonomous driving cars and industry. In such cases, knowing the training data labels and semantics, such as *"healthy CT scan images"*, is possible and we do not need more details about normal data classes except for their names.

Moreover, previous adversarially robust OOD detector models have reported almost no improvement over random results in real-world datasets, especially in the case of ND settings. Therefore, our use of the normal class label as an alternative solution is reasonable. Our approach outperforms previous models by up to 50% in the robust ND scenario and this superiority continues in multi-class modes where data labels are available and we only use the class names to improve the model. Given the applicability of the task addressed in this article and the progress of multi-domain models, our approach has potential for practical use

## 14    LEVERAGING PRE-TRAINED MODELS FOR OOD DETECTION

It has been demonstrated that leveraging pre-trained models can significantly improve the performance and practical applicability of downstream tasks (89; 90), including OOD detection, which has been extensively studied.

Various works (47; 48; 49; 11; 51; 91) have utilized pre-trained models' features or transfer learning techniques to improve detection results and efficiency, particularly in OOD detection under harder constraints. For example, (88) used a pre-trained CLIP model trained on 400 million data for Zero-Shot OOD Detection, (92) proposed using a pre-trained ViT (72) model trained on 22 million data for near-distribution OOD detection, and (93) utilized a pre-trained BERT (94) model trained on billions of data for OOD detection in the text domain. In our work, we addressed the highly challenging task of developing an adversarially robust OOD detector model, which is unexplored for real-world datasets such as medical datasets. To accomplish this, we utilized the CLIP and diffusion model as our generator backbone, which was trained on 67 million data.

## 15 WHY OUR DIFFUSION MODEL IS THE BEST FIT FOR NEAR-OOD GENERATION

**Working in Pixel Space**   In Section 18.1, we discussed how OOD data can be divided into two categories (i.e. pixel- and semantic-level). Our need for diverse OOD data motivates our preference for generative models that can create both pixel-level and semantic-level OOD data. Our generative model is a suitable choice as it uses a diffusion model applied at the pixel-level to generate images from texts. This allows the model to generate OOD samples that differ in their local appearance, which is particularly important for pixel-level OOD detection. Compared to other SOTA text-to-image models that mostly work at the embedding level, our generative model's ability to generate images at the pixel-level makes it a better choice for our purposes.

**Comparing with DreamBooth**   As our pipeline's generator model involves image editing, we explored the literature on image manipulating and tested a common methods used for image editing. Numerous algorithms have been proposed for generating new images conditioned on input images among these, we have chosen DreamBooth as one of the SOTA algorithms for specifying image details in text-to-image models. we evaluated the DreamBooth algorithm for changing image details in various datasets. Our experiment showed that, despite DreamBooth's good performance for natural images and human faces, the algorithm had poor results for datasets with different distributions, such as MNIST and FashionMNIST. One possible explanation for the poor performance of these algorithms is their bias towards the distribution of the training datasets, such as LAION, which typically consists of natural images and portraits. Consequently, these algorithms may not yield satisfactory results for datasets with different distributions.

**Inliers**                    **Generated with Stable Diffusion + DreamBooth**

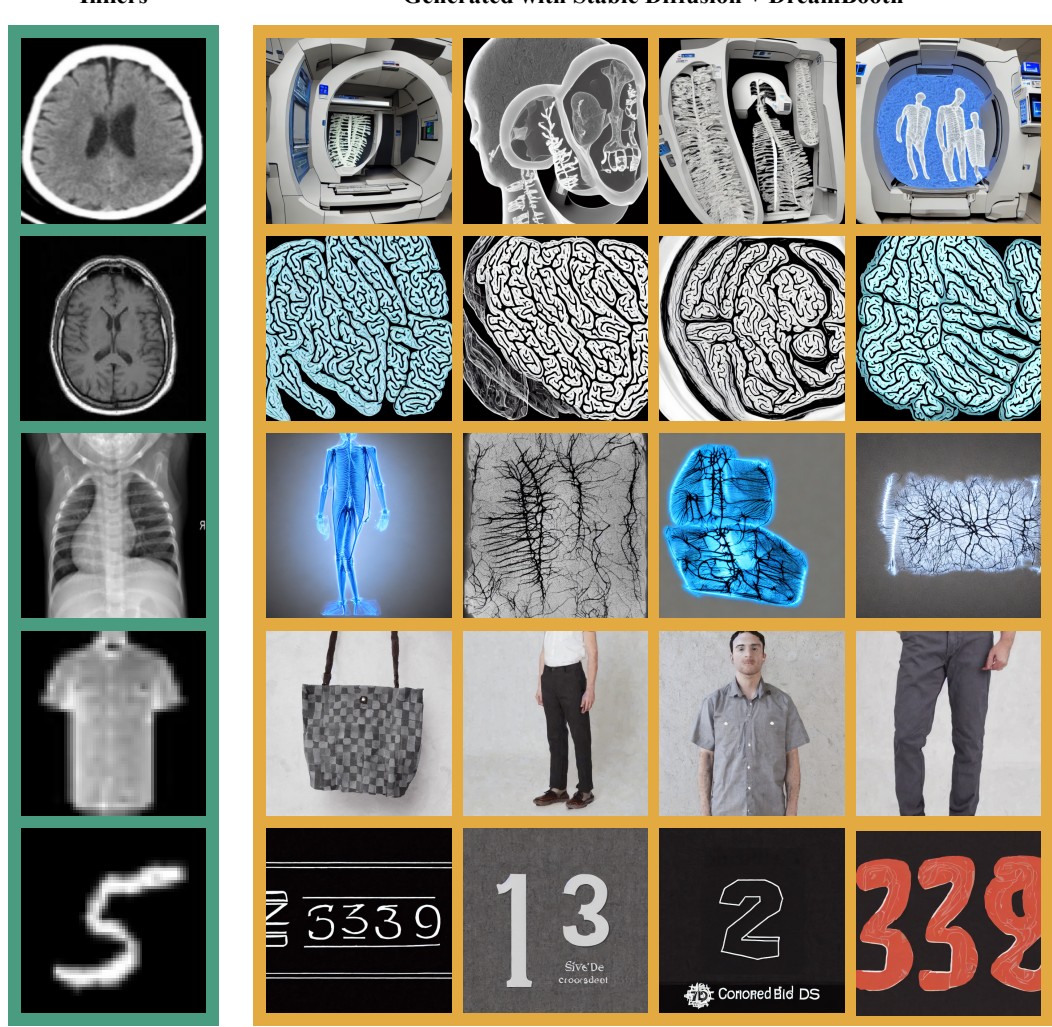

Figure 7: generated images using the DreamBooth algorithm and StableDiffusion model, which shows a very large shift between ID- and generated OOD data. This demonstrates the superiority of our pipeline as a near OOD generator.

# 16  DETAILED ANALYSIS AND INSIGHTS OF ABLATION STUDY

**FID, Density and Coverage**  Fréchet Inception Distance (FID) metric (67) measures the distance between feature vectors of real and generated images and calculates the distance them, has been shown to match with human judgments. The diversity of generative models can be evaluated using two metrics: Density and Coverage (68). By utilizing a manifold estimation procedure code, the distance between two sets of images can be measured. To calculate these metrics, features from a pre-trained model are utilized, specifically those before the final classification layer. The mathematical expression for these metrics is as follows:

$$Density\left(\boldsymbol{X}_s, \boldsymbol{X}_t, F, k\right) = \frac{1}{kM} \sum_{j=1}^{M} \sum_{i=1}^{N} \mathbb{I}\left(\boldsymbol{f}_{t,j} \in B\left(\boldsymbol{f}_{s,i}, \mathrm{NN}_k\left(F\left(\boldsymbol{X}_s\right), \boldsymbol{f}_{s,i}, k\right)\right)\right), \tag{18}$$

$$Coverage\left(\boldsymbol{X}_s, \boldsymbol{X}_t, F, k\right) = \frac{1}{N} \sum_{i=1}^{N} \mathbb{I}\left(\exists j \text{ s.t. } \boldsymbol{f}_{t,j} \in B\left(\boldsymbol{f}_{s,i}, \mathrm{NN}_k\left(F\left(\boldsymbol{X}_s\right), \boldsymbol{f}_{s,i}, k\right)\right)\right). \tag{19}$$

where $\boldsymbol{F}$ is a feature extractor, $\boldsymbol{f}$ is a collection of features from $F$,

$\boldsymbol{X}_s = \{\boldsymbol{x}_{s,1}, \dots, \boldsymbol{x}_{s,N}\}$ denotes real images, $\boldsymbol{X}_t = \{\boldsymbol{x}_{t,1}, \dots, \boldsymbol{x}_{t,M}\}$ denotes generated images, $B(\boldsymbol{f}, r)$ is the n-dimensional sphere in which $\boldsymbol{f} = F(\boldsymbol{x}) \in \mathbb{R}^n$ is the center and $r$ is the radius, $\mathrm{NN}_k f(F, \boldsymbol{f}, k)$ is the

distance from $f$ to the $k$-th nearest embedding in $F$, and $\mathbb{I}(\cdot)$ is a indicator function. We use the standard InceptionV3 features, which are also used to compute the FID. The measures are computed using the official code (68).

The definition of the FDC metric introduced in the paper is as below:

$$FDC = \frac{\log(FID^{-1})}{\log(Density) + \log(Coverage)} \tag{20}$$

## 17 The Significance of Conditioning on Both Images and Text from the Normal Distribution

In order to have an accurate OOD detector, it's important to generate diverse and realistic samples that are close to the distribution of the normal data. In our study, we tackle this challenge by leveraging the information contained in the normal data. Specifically, we extract the labels of classes that are close to the normal set and use them as guidance for generation. Additionally, we initialize the reverse process generation of a diffusion model with normal images, so the generation of OOD data in our pipeline is conditioned on both the images and the text of the normal set. This enables us to generate adaptive OOD samples.

In the Ablation Study (sec. 5), we demonstrate the importance of using both image and text information for generating OOD data. We compare our approach with two other methods that condition on only one type of information and ignore the other. The first technique generates fake images based on the normal set, while the other generates OOD data using only the extracted text from normal labels. The results show that both techniques are less effective than our adaptive exposure technique, which conditions the generation process on both text and image. This confirms that using both sources of information is mandatory and highly beneficial.

### 17.1 Samples Generated Solely Based on Text Conditioning

In this section, we compare normal images with images generated by our pipline using only text in Fig. 8 (without conditioning on the images). Our results, illustrated by the plotted samples, demonstrate that there is a significant difference in distribution between these generated images and normal images. This difference is likely the reason for the ineffectiveness of the OOD samples generated with this technique.

## 18 Label Generation

### 18.1 Pixel-Level and Semantic-Level OOD Detection

OOD samples can be categorized into two types: pixel-level and semantic-level. In pixel-level OOD detection, ID and OOD samples differ in their local appearance, while remaining semantically identical. For instance, a broken glass could be considered an OOD sample compared to an intact glass due to its different local appearance. In contrast, semantic-level OOD samples differ at the semantic level, meaning that they have different meanings or concepts than the ID samples. For example, a cat is an OOD sample when we consider dog semantics as ID because they represent different concepts.

### 18.2 Our Method of Generating labels

A reliable and generalized approach for anomaly detection must have the capability to detect both semantic-level and pixel-wise anomalies, as discussed in the previous section. To this end, our proposed method constructs NOOD labels by combining two sets of words: near-distribution labels and negative adjectives derived from a normal label name. We hypothesize that the former set can detect semantic-level anomalies, while the latter set is effective in detecting pixel-wise anomalies. Additionally, we include an extra labels, marked as 'others', in the labels list to augment the diversity of exposures.

To generate negative adjectives, we employ a set of constant texts that are listed below and used across all experimental settings (X is the normal label name):

- A photo of X with a crack
- A photo of a broken X
- A photo of X with a defect
- A photo of X with damage
- A photo of X with a scratch
- A photo of X with a hole

- A photo of X torn
- A photo of X cut
- A photo of X with contamination
- A photo of X with a fracture
- A photo of a damaged X
- A photo of a fractured X
- A photo of X with destruction
- A photo of X with a mark

For NOOD labels, we utilize Word2Vec to search for semantically meaningful word embeddings after normalizing the words through a process of lemmatization. First, we obtain the embedding of the normal class label and then search among the corpus to identify the 1000 nearest neighbors of the normal class label.

In the subsequent phase, we employ the combination of Imagenet labels and CLIP to effectively identify and eliminate labels that demonstrate semantic equivalence to the normal label. Initially, we leverage CLIP to derive meaningful representations of the Imagenet labels. Then, we calculate the norm of the pairwise differences among these obtained representations. By computing the average of these values, a threshold is established, serving as a determinant of the degree of semantic similarity between candidate labels and the normal label. The threshold is defined as:

$$\tau_{label} = \frac{\sum_{i=1, i \neq j}^{M} \sum_{j=1}^{M} |E_T(y_i) - E_T(y_j)|}{M(M-1)} \tag{21}$$

In which, $M$ is the number of Imagenet labels, and $y_i$s are the Imagenet labels.

Consequently, we filter out labels whose CLIP output exhibits a discrepancy from the normal class that falls below the threshold.

We then sample NOOD labels from the obtained words based on the similarity factor of the neighbors to the normal class label. The selection probability of the NOOD labels is proportional to their similarity to the normal class label. Finally, we compile a list of NOOD labels to serve as near OOD labels.

## 19 OSR Experiments Details

In order to evaluate earlier works in OSR setting, we first select desired number of classes, say $K$ and rename the labels of samples to be in the range 0 to $K - 1$. Then following the guideline of the method, we evaluate it in both clean and adversarial settings and repeat each experiment 5 times and report the average.

## 20 OOD Experiments Details

Table 2 yielded results that are now presented in Table 7 for a more comprehensive overview. We designated multiple datasets as out-of-distribution during the testing phase and reported the outcomes in Table 7. Adversarial and clean out-of-distribution scenarios have also been examined by other approaches. Prominent methods in the clean setting encompass the ViT architecture and OpenGAN. Regarding image classification, AT and HAT have been recognized as highly effective defenses. AOE, ALOE, and OSAD are regarded as state-of-the-art methods for out-of-distribution detection, and ATD in robust OOD detection. These OOD methods (excluding OpenGAN and ATD) have undergone evaluation with various detection techniques, including MSP (95)(96), MD (97), Relative MD (98), and OpenMax (99). The results reported for each OOD method correspond to the best-performing detection method. Notably, our approach has surpassed the state-of-the-art in robust out-of-distribution setting (ATD) for nearly all datasets.

$$\mu_k = \frac{1}{N} \sum_{i: y_i = k} z_i, \quad \Sigma = \frac{1}{N} \sum_{k=1}^{K} \sum_{i: y_i = k} (z_i - \mu_k)(z_i - \mu_k)^T, \quad k = 1, 2, \ldots, K \tag{22}$$

In addition, to use RMD, one has to fit a $\mathcal{N}(\mu_0, \Sigma_0)$ to the whole in-distribution. Next, the distances and anomaly score for the input $x'$ with pre-logits $z'$ are computed as:

$$MD_k(z') = (z' - \mu_k)^T \Sigma^{-1}(z' - \mu_k), \quad RMD_k(z') = MD_k(z') - MD_0(z'),$$
$$\text{score}_{MD}(x') = -\min_k \{MD_k(z')\}, \quad \text{score}_{RMD}(x') = -\min_k \{RMD_k(z')\}. \tag{23}$$

Table 6: The detailed AUROC scores of the class-specific experiments for (One-Class) Novelty Detection setting in CIFAR10, CIFAR100, MNIST, Fashion-MNIST datasets.

(a) CIFAR10

| Method | Attack | Class | | | | | | | | | | Average |
|--------|--------|------|------|------|------|------|------|------|------|------|------|---------|
| | | 0 | 1 | 2 | 3 | 4 | 5 | 6 | 7 | 8 | 9 | |
| Ours | Clean | 91.7 | 97.3 | 77.4 | 74.0 | 82.6 | 81.2 | 91.5 | 92.8 | 94.0 | 91.7 | 87.4 |
| | BlackBox | 89.9 | 95.8 | 75.5 | 72.1 | 81.6 | 79.1 | 89.1 | 91.2 | 92.4 | 89.6 | 85.6 |
| | PGD-100 | 78.0 | 82.1 | 59.9 | 55.3 | 65.4 | 67.5 | 74.3 | 70.2 | 80.2 | 77.8 | 71.1 |
| | AutoAttack | 75.7 | 80.0 | 58.7 | 53.6 | 64.2 | 65.1 | 72.1 | 68.8 | 78.9 | 76.0 | 69.3 |

(b) CIFAR100

| Method | Attack | Class | | | | | | | | | | | | | | | | | | | | Average |
|--------|--------|------|------|------|------|------|------|------|------|------|------|------|------|------|------|------|------|------|------|------|------|---------|
| | | 0 | 1 | 2 | 3 | 4 | 5 | 6 | 7 | 8 | 9 | 10 | 11 | 12 | 13 | 14 | 15 | 16 | 17 | 18 | 19 | |
| Ours | Clean | 79.0 | 78.6 | 95.5 | 78.1 | 89.2 | 69.4 | 76.2 | 82.6 | 77.9 | 87.4 | 92.7 | 73.1 | 77.6 | 65.0 | 83.9 | 65.5 | 72.1 | 93.6 | 83.5 | 72.0 | 79.6 |
| | BlackBox | 76.9 | 77.0 | 93.2 | 75.4 | 87.6 | 67.3 | 74.0 | 79.6 | 75.7 | 85.1 | 89.9 | 71.3 | 74.7 | 62.9 | 81.0 | 63.4 | 69.7 | 91.8 | 81.8 | 70.1 | 77.4 |
| | PGD-100 | 59.9 | 61.2 | 83.2 | 62.5 | 74.2 | 53.9 | 61.6 | 63.0 | 59.2 | 76.9 | 78.4 | 51.7 | 61.3 | 49.0 | 61.0 | 45.0 | 54.3 | 79.3 | 61.5 | 59.2 | 62.8 |
| | AutoAttack | 56.8 | 59.4 | 80.4 | 60.9 | 73.9 | 51.2 | 61.3 | 61.7 | 58.3 | 75.9 | 75.7 | 51.7 | 59.9 | 45.7 | 59.2 | 44.4 | 50.6 | 76.1 | 60.5 | 57.1 | 61.0 |

(c) MNIST

| Method | Attack | Class | | | | | | | | | | Average |
|--------|--------|------|------|------|------|------|------|------|------|------|------|---------|
| | | 0 | 1 | 2 | 3 | 4 | 5 | 6 | 7 | 8 | 9 | |
| Ours | Clean | 99.8 | 99.4 | 99.3 | 99.2 | 99.6 | 99.4 | 99.8 | 98.9 | 99.4 | 98.8 | 99.4 |
| | BlackBox | 98.7 | 99.0 | 98.2 | 98.8 | 98.3 | 98.9 | 99.4 | 97.8 | 98.5 | 98.2 | 98.6 |
| | PGD-100 | 97.4 | 96.4 | 97.0 | 92.9 | 97.5 | 96.6 | 98.4 | 93.2 | 95.4 | 92.6 | 95.7 |
| | AutoAttack | 96.9 | 96.1 | 96.3 | 92.0 | 96.7 | 96.3 | 98.1 | 92.5 | 95.2 | 92.0 | 95.2 |

(d) Fashion-MNIST

| Method | Attack | Class | | | | | | | | | | Average |
|--------|--------|------|------|------|------|------|------|------|------|------|------|---------|
| | | 0 | 1 | 2 | 3 | 4 | 5 | 6 | 7 | 8 | 9 | |
| Ours | Clean | 95.8 | 99.7 | 93.9 | 93.4 | 92.9 | 98.3 | 86.5 | 98.6 | 98.5 | 98.8 | 95.6 |
| | BlackBox | 94.4 | 98.6 | 92.7 | 92.6 | 91.2 | 96.9 | 85.1 | 97.1 | 97.5 | 97.1 | 94.3 |
| | PGD-100 | 90.0 | 98.5 | 83.8 | 81.1 | 77.5 | 95.3 | 72.7 | 95.1 | 92.5 | 94.5 | 88.1 |
| | AutoAttack | 89.7 | 98.1 | 83.0 | 80.9 | 76.5 | 95.1 | 72.6 | 94.2 | 92.4 | 94.1 | 87.6 |

Table 7: OOD detailed results

(a) CIFAR10

| Out-Dataset | Attack | Method | | | | | | | |
|---|---|---|---|---|---|---|---|---|---|
| | | OpenGAN | ViT (RMD) | ATOM | AT (OpenMax) | OSAD (OpenMax) | ALOE (MSP) | ATD | RODEO |
| MNIST | Clean | 99.4 | 98.7 | 98.4 | 80.4 | 86.2 | 74.6 | 98.8 | 96.9 |
| | PGD-100 | 30.3 | 3.5 | 0.0 | 39.6 | 55.3 | 22.7 | **90.2** | 84.0 |
| TiImgNet | Clean | 95.3 | 95.2 | 97.2 | 81.0 | 81.9 | 82.1 | 88.0 | 85.1 |
| | PGD-100 | 15.2 | 2.3 | 4.3 | 16.5 | 19.3 | 21.6 | 47.0 | 47.2 |
| Places | Clean | 95.0 | **98.3** | 98.7 | 82.5 | 83.3 | 85.1 | 92.5 | 96.2 |
| | PGD-100 | 17.3 | 3.1 | 6.5 | 18.9 | 21.2 | 22.8 | 60.7 | **71.1** |
| LSUN | Clean | 96.5 | 98.4 | 99.1 | 85.0 | 86.4 | 98.7 | 96.0 | **99.0** |
| | PGD-100 | 24.0 | 2.0 | 1.9 | 19.6 | 20.7 | 51.6 | 69.0 | **86.0** |
| iSUN | Clean | 96.3 | **98.6** | 99.5 | 83.9 | 84.0 | 98.3 | 94.8 | 97.7 |
| | PGD-100 | 23.0 | 2.1 | 3.4 | 19.5 | 20.3 | 50.4 | 66.8 | **79.6** |
| Birds | Clean | **98.3** | 76.0 | 95.8 | 75.1 | 76.5 | 79.9 | 93.6 | 97.8 |
| | PGD-100 | 34.5 | 0.6 | 6.1 | 14.7 | 19.1 | 21.8 | 69.0 | 76.9 |
| Flower | Clean | 98.3 | 99.6 | 99.8 | 85.5 | 88.6 | 79.0 | **99.7** | 99.5 |
| | PGD-100 | 30.1 | 2.6 | 19.9 | 20.9 | 26.6 | 19.6 | **93.7** | 89.6 |
| COIL | Clean | **98.1** | 95.9 | 97.3 | 70.3 | 75.0 | 76.8 | 90.8 | 91.1 |
| | PGD-100 | 38.5 | 3.9 | 9.5 | 16.6 | 18.7 | 19.3 | 58.1 | **60.4** |
| CIFAR100 | Clean | 95.0 | **97.3** | 94.2 | 79.6 | 79.9 | 78.8 | 82.0 | 75.6 |
| | PGD-100 | 10.1 | 1.70 | 2.5 | 16.0 | 18.1 | 17.0 | 38.0 | **38.7** |
| Avg. | Clean | **97.1** | 95.1 | 97.8 | 80.5 | 82.7 | 84.3 | 94.3 | 93.2 |
| | PGD-100 | 26.6 | 2.5 | 6.0 | 20.8 | 25.1 | 28.7 | 69.3 | **70.4** |

(b) CIFAR100

| Out-Dataset | Attack | Method | | | | | | | |
|---|---|---|---|---|---|---|---|---|---|
| | | OpenGAN | ViT (RMD) | ATOM | AT (RMD) | OSAD (MD) | ALOE(MD) | ATD | RODEO |
| MNIST | Clean | 99.0 | 83.8 | 90.4 | 41.1 | 95.9 | 96.6 | 97.3 | **99.7** |
| | PGD-100 | 14.6 | 1.7 | 0.0 | 14.2 | 82.0 | 73.1 | 86.3 | **97.7** |
| TiImgNet | Clean | 88.3 | **90.1** | 85.1 | 72.3 | 48.3 | 58.1 | 73.7 | 72.9 |
| | PGD-100 | 3.9 | 3.1 | 1.8 | 12.0 | 9.9 | 6.3 | 26.0 | **39.0** |
| Places | Clean | **94.5** | 92.3 | 94.8 | 73.1 | 55.7 | 75.0 | 83.3 | 93.0 |
| | PGD-100 | 4.9 | 3.7 | 4.7 | 12.7 | 12.1 | 14.1 | 41.7 | **68.3** |
| LSUN | Clean | 97.1 | 91.6 | 96.6 | 76.0 | 55.6 | 83.1 | 89.2 | **98.1** |
| | PGD-100 | 7.3 | 0.6 | 3.2 | 12.9 | 10.4 | 20.7 | 49.4 | **84.8** |
| iSUN | Clean | **96.4** | 91.4 | 96.4 | 72.5 | 54.8 | 80.1 | 86.5 | 95.1 |
| | PGD-100 | 7.5 | 0.9 | 3.1 | 11.9 | 10.6 | 22.1 | 47.3 | **77.3** |
| Birds | Clean | 96.6 | **97.8** | 95.1 | 73.1 | 54.5 | 78.4 | 93.4 | 96.8 |
| | PGD-100 | 7.4 | 10.5 | 14.2 | 13.4 | 11.0 | 23.7 | 66.2 | 75.9 |
| Flower | Clean | 96.8 | 96.6 | 98.9 | 77.6 | 69.6 | 85.1 | **97.2** | 97.2 |
| | PGD-100 | 9.3 | 5.5 | 17.2 | 15.7 | 22.9 | 31.8 | **80.1** | 78.9 |
| COIL | Clean | **97.7** | 88.1 | 79.5 | 74.4 | 57.5 | 77.9 | 80.6 | 78.6 |
| | PGD-100 | 15.7 | 3.5 | 0.8 | 16.3 | 14.0 | 19.2 | **45.3** | 44.8 |
| CIFAR10 | Clean | 92.9 | **94.8** | 87.5 | 67.5 | 50.3 | 43.6 | 57.5 | 61.5 |
| | PGD-100 | 9.1 | 5.8 | 3.7 | 10.7 | 10.3 | 3.0 | 13.8 | **30.7** |
| Avg. | Clean | **95.8** | 91.5 | 91.6 | 70.0 | 61.5 | 79.3 | 87.7 | 88.1 |
| | PGD-100 | 8.8 | 3.7 | 5.4 | 13.6 | 21.6 | 26.4 | 55.3 | **66.4** |

Table 8: AUROC scores for (One-Class) Novelty Detection under three different adversarial attacks with (a) $\epsilon = \frac{8}{255}$ and (b) $\epsilon = \frac{2}{255}$. The best and second-best results are highlighted in bold and underlined format respectively in each row.

| | Dataset | Attack | DeepSVDD | CSI | DN2 | PANDA | MSAD | Transformaly | PatchCore | PrincipaLS | OCSDF | APAE | RODEO |
|---|---|---|---|---|---|---|---|---|---|---|---|---|---|
| **Low-Res** | CIFAR10 | Clean | 64.8 | 94.3 | 92.5 | 96.2 | 97.2 | **98.3** | 68.3 | 57.7 | 57.1 | 55.2 | 87.4 |
| | | BlackBox | 54.6 | 43.1 | 31.8 | 45.3 | 38.4 | 62.9 | 18.1 | 33.3 | 48.4 | 27.6 | **85.6** |
| | | PGD-100 | 23.4 | 2.8 | 0.0 | 0.0 | 0.0 | 0.0 | 0.3 | 24.1 | 22.0 | 0.1 | **71.1** |
| | | AutoAttack | 9.7 | 0.0 | 0.5 | 0.0 | 0.0 | 0.0 | 0.0 | 20.2 | 15.3 | 0.0 | **69.3** |
| | CIFAR100 | Clean | 67.0 | 89.6 | 89.3 | 94.1 | 96.4 | **97.3** | 66.8 | 52.0 | 48.2 | 51.8 | 79.6 |
| | | BlackBox | 55.3 | 34.7 | 28.5 | 42.6 | 51.8 | 64.0 | 23.6 | 29.4 | 36.9 | 16.3 | **77.4** |
| | | PGD-100 | 14.5 | 3.2 | 0.0 | 1.6 | 2.9 | 4.3 | 0.0 | 16.6 | 15.1 | 0.0 | **62.8** |
| | | AutoAttack | 5.8 | 0.0 | 0.0 | 0.0 | 4.3 | 2.6 | 0.0 | 14.7 | 12.0 | 0.0 | **61.0** |
| | MNIST | Clean | 94.8 | 93.8 | 95.7 | 98.0 | 96.0 | 94.8 | 83.2 | 97.3 | 95.5 | 92.5 | **99.4** |
| | | BlackBox | 65.7 | 72.3 | 56.4 | 62.2 | 58.1 | 73.5 | 46.9 | 80.6 | 75.7 | 73.0 | **98.6** |
| | | PGD-100 | 11.4 | 0.3 | 0.0 | 5.3 | 0.0 | 10.8 | 0.0 | 78.1 | 62.4 | 22.9 | **95.7** |
| | | AutoAttack | 9.6 | 0.4 | 0.0 | 0.6 | 7.3 | 6.7 | 0.0 | 72.5 | 58.3 | 19.8 | **95.2** |
| | Fashion-MNIST | Clean | 94.5 | 92.7 | 94.4 | **95.6** | 94.2 | 94.4 | 77.4 | 91.0 | 90.6 | 86.1 | 95.6 |
| | | BlackBox | 66.8 | 64.2 | 42.5 | 53.1 | 73.8 | 79.6 | 58.2 | 71.1 | 67.0 | 24.3 | **94.3** |
| | | PGD-100 | 49.4 | 4.8 | 0.0 | 0.0 | 0.0 | 0.5 | 0.0 | 61.3 | 54.5 | 11.2 | **88.1** |
| | | AutoAttack | 38.2 | 3.1 | 0.0 | 4.9 | 0.0 | 0.0 | 0.0 | 58.2 | 49.2 | 7.0 | **87.6** |
| **High-Res** | MVTecAD | Clean | 67.0 | 63.6 | 81.4 | 86.5 | 87.2 | 87.9 | **99.6** | 63.8 | 58.7 | 62.1 | 61.5 |
| | | BlackBox | 36.0 | 37.7 | 48.5 | 46.9 | 41.3 | 56.0 | 58.3 | 45.2 | 33.4 | 35.9 | **60.0** |
| | | PGD-100 | 2.6 | 0.0 | 0.0 | 5.8 | 0.9 | 0.0 | 7.2 | **24.3** | 5.2 | 4.7 | 15.9 |
| | | AutoAttack | 0.0 | 0.0 | 0.0 | 0.7 | 0.0 | 0.0 | 4.8 | 12.6 | 0.3 | 1.8 | **14.2** |
| | Head-CT | Clean | 62.5 | 60.9 | 64.0 | 64.5 | 59.4 | 78.1 | **98.5** | 68.9 | 62.4 | 68.1 | 87.3 |
| | | BlackBox | 44.1 | 50.3 | 52.1 | 48.7 | 42.6 | 65.0 | 80.7 | 54.3 | 40.2 | 45.2 | **85.6** |
| | | PGD-100 | 0.0 | 0.6 | 0.7 | 0.0 | 0.0 | 6.4 | 1.5 | 27.8 | 13.1 | 6.6 | **70.0** |
| | | AutoAttack | 0.0 | 0.0 | 0.0 | 0.5 | 0.0 | 3.2 | 0.0 | 16.2 | 8.5 | 3.8 | **68.4** |
| | BrainMRI | Clean | 74.5 | 93.2 | 67.6 | 72.5 | **99.9** | 98.3 | 91.4 | 70.2 | 63.2 | 55.4 | 76.3 |
| | | BlackBox | 52.7 | 61.0 | 14.7 | 8.1 | 64.2 | 71.6 | 72.5 | 56.9 | 48.0 | 27.1 | **75.8** |
| | | PGD-100 | 4.3 | 0.0 | 2.9 | 0.0 | 1.7 | 5.2 | 0.0 | 33.5 | 20.4 | 9.7 | **71.1** |
| | | AutoAttack | 2.1 | 0.0 | 0.0 | 0.0 | 0.0 | 1.6 | 0.4 | 17.8 | 12.5 | 8.3 | **70.5** |
| | Tumor Detection | Clean | 70.8 | 85.3 | 71.1 | 75.3 | 95.1 | **97.4** | 92.8 | 73.5 | 65.2 | 64.6 | 89.0 |
| | | BlackBox | 42.0 | 60.9 | 54.7 | 58.2 | 67.7 | 78.6 | 67.2 | 56.4 | 35.0 | 43.1 | **87.2** |
| | | PGD-100 | 1.7 | 0.0 | 0.4 | 0.0 | 0.1 | 7.4 | 9.3 | 25.2 | 17.9 | 15.8 | **67.5** |
| | | AutoAttack | 0.0 | 2.2 | 0.0 | 0.3 | 0.0 | 5.1 | 6.1 | 14.7 | 10.1 | 8.3 | **66.9** |
| | Covid19 | Clean | 61.9 | 65.1 | 88.5 | 76.4 | 89.2 | 91.0 | 77.7 | 54.2 | 46.1 | 50.7 | **79.6** |
| | | BlackBox | 32.4 | 25.7 | 43.2 | 30.0 | 53.6 | 70.7 | 56.3 | 43.8 | 28.5 | 26.1 | **75.0** |
| | | PGD-100 | 0.0 | 0.0 | 0.6 | 1.5 | 4.7 | 10.6 | 4.2 | 15.3 | 9.0 | 11.2 | **59.4** |
| | | AutoAttack | 0.0 | 0.3 | 0.0 | 0.2 | 1.9 | 4.4 | 0.5 | 9.1 | 6.5 | 8.7 | **58.8** |

Table 9: Detailed comparison of different exposure techniques and our introduced method of Adaptive Exposure over different datasets.

(a) AUROC(%) for different exposure techniques

| Exposure Technique / Target Dataset | Attack | CIFAR10 | CIFAR100 | MNIST | Fashion-MNIST | MVTec-ad | Head-CT | Brain-MRI | Tumor Detection | Covid19 |
|---|---|---|---|---|---|---|---|---|---|---|
| Gaussian Noise | Clean | 64.4 | 54.6 | 60.1 | 62.7 | 41.9 | 59.0 | 45.3 | 51.7 | 40.7 |
| | PGD | 15.3 | 12.0 | 12.4 | 15.7 | 0.1 | 0.8 | 0.1 | 1.4 | 0.0 |
| ImageNet (Fixed OE Dataset) | Clean | 87.3 | **79.6** | 90.0 | 93.0 | **64.6** | 61.8 | 69.3 | 71.8 | 62.7 |
| | PGD | 70.0 | **64.8** | 43.0 | 82.0 | 0.5 | 2.1 | 1.1 | 23.0 | 24.5 |
| Mixup with ImageNet | Clean | 59.4 | 56.1 | 59.6 | 74.2 | 58.5 | 54.4 | 57.3 | 76.4 | 69.2 |
| | PGD | 31.5 | 27.3 | 1.7 | 48.8 | 1.4 | 21.4 | 12.1 | 53.5 | 50.8 |
| Fake Image Generation | Clean | 29.5 | 23.0 | 76.0 | 52.2 | 43.5 | 63.7 | 65.2 | 65.2 | 42.7 |
| | PGD | 16.2 | 14.8 | 51.3 | 31.1 | 7.3 | 6.9 | 28.8 | 32.7 | 13.0 |
| Stable Diffusion Prompt | Clean | 62.4 | 54.8 | 84.3 | 63.7 | 54.9 | 71.5 | 66.7 | 45.8 | 37.1 |
| | PGD | 35.6 | 35.7 | 62.5 | 48.5 | 12.6 | 3.6 | 7.4 | 5.6 | 0.0 |
| **Adaptive Exposure** | Clean | **87.4** | **79.6** | **99.4** | **95.6** | 61.5 | **87.3** | **76.3** | **89.0** | **79.6** |
| | PGD | **71.1** | 62.8 | **95.7** | **88.1** | **15.9** | **70.0** | **71.1** | **67.5** | **59.4** |

(b) FID, Density, and Coverage metrics for different exposure techniques

| Exposure Technique / Target Dataset | Metric | CIFAR10 | CIFAR100 | MNIST | Fashion-MNIST | MVTec-ad | Head-CT | Brain-MRI | Tumor Detection | Covid19 |
|---|---|---|---|---|---|---|---|---|---|---|
| Gaussian Noise | FID | 340 | 326 | 407 | 399 | 493 | 510 | 503 | 498 | 501 |
| | D / C | 0.04 / 0.02 | 0.03 / 0.02 | 0.01 / 0.02 | 0.03 / 0.02 | 0.04 / 0.01 | 0.02 / 0.01 | 0.03 / 0.01 | 0.05 / 0.03 | 0.01 / 0.01 |
| ImageNet (Fixed OE Dataset) | FID | 210 | 236 | 365 | 337 | 362 | 330 | 395 | 320 | 460 |
| | D / C | 0.71 / 0.70 | 0.71 / 0.64 | 0.02 / 0.30 | 0.44 / 0.21 | 0.21 / 0.23 | 0.36 / 0.43 | 0.22 / 0.08 | 0.32 / 0.17 | 0.43 / 0.48 |
| Mixup with ImageNet | FID | 57 | 59 | 161 | 110.9 | 295 | 204 | 275 | 225 | 360 |
| | D / C | 0.37 / 0.42 | 0.44 / 0.39 | 0.01 / 0.05 | 0.08 / 0.11 | 0.27 / 0.17 | 0.27 / 0.13 | 0.15 / 0.28 | 0.23 / 0.34 | 0.31 / 0.19 |
| Fake Image Generation | FID | 98 | 105 | 193 | 256 | 448 | 432 | 294 | 358 | 415 |
| | D / C | 0.08 / 0.12 | 0.06 / 0.14 | 0.01 / 0.03 | 0.01 / 0.17 | 0.13 / 0.20 | 0.02 / 0.06 | 0.29 / 0.16 | 0.25 / 0.18 | 0.04 / 0.03 |
| Stable Diffusion Prompt | FID | 212 | 138 | 228 | 272 | 395 | 349 | 240 | 211 | 368 |
| | D / C | 0.34 / 0.05 | 0.29 / 0.10 | 0.28 / 0.17 | 0.11 / 0.18 | 0.37 / 0.14 | 0.08 / 0.06 | 0.07 / 0.41 | 0.25 / 0.07 | 0.01 / 0.04 |
| **Adaptive Exposure** | FID | 145 | 156 | 133 | 134 | 263 | 204 | 165 | 186 | 201 |
| | D / C | 0.87 / 0.64 | 0.63 / 0.62 | 0.75 / 0.86 | 0.61 / 0.44 | 0.64 / 0.09 | 0.77 / 0.83 | 0.69 / 0.61 | 0.57 / 0.37 | 0.51 / 0.80 |

Table 10: The detailed AUROC scores of RODEO for (One-Class) Novelty Detection setting under the different adversarial attacks.

| Attack | CIFAR10 | CIFAR100 | MNIST | FashionMNIST | SVHN | MVTecAD | Head-CT | BrainMRI | Tumor Detection | Covid19 |
|---|---|---|---|---|---|---|---|---|---|---|
| Clean | 87.4 | 79.6 | 99.4 | 95.6 | 75.2 | 61.5 | 87.3 | 76.3 | 89.0 | 79.6 |
| BlackBox | 85.6 | 77.4 | 98.6 | 94.3 | 74.7 | 60.0 | 85.6 | 75.8 | 87.2 | 75.0 |
| PGD-100 | 71.1 | 62.8 | 95.7 | 88.1 | 35.4 | 15.9 | 70.0 | 71.1 | 67.5 | 59.4 |
| PGD-1000 * | 70.2 | 62.1 | 94.6 | 87.2 | 33.8 | 14.9 | 68.6 | 68.4 | 67.0 | 58.3 |
| AutoAttack | 69.3 | 61.0 | 95.2 | 87.6 | 33.2 | 14.2 | 68.4 | 70.5 | 66.9 | 58.8 |
| Adaptive Auto Attack | 70.5 | 61.3 | 94.0 | 87.0 | 31.8 | 13.4 | 68.1 | 67.7 | 65.6 | 57.6 |

Table 11: The detailed AUROC scores of Open-Set Recognition (OSR) setting under the different adversarial attacks.

| Attack | CIFAR10 | CIFAR100 | MNIST | FashionMNIST |
|---|---|---|---|---|
| Clean | 97.2 | 88.8 | 79.6 | 48.5 |
| PGD-100 | 87.7 | 67.1 | 64.1 | 37.7 |
| PGD-1000 | 85.0 | 65.3 | 62.7 | 35.3 |
| AutoAttack | 86.4 | 66.8 | 63.5 | 36.9 |
| AdaptiveAutoAttack | 84.1 | 62.9 | 63.0 | 35.4 |

Table 12: The detailed AUROC scores of RODEO for OOD Detection setting under the different adversarial attacks.

| Dataset | Attack | | | | |
|---|---|---|---|---|---|
| | Clean | PGD-100 | PGD-1000 | AutoAttack | Adaptive Auto Attack |
| CIFAR10 | 93.2 | 70.4 | 69.5 | 69.0 | 68.8 |
| CIFAR100 | 88.1 | 66.4 | 64.7 | 65.3 | 63.2 |

Table 13: The detailed AUROC scores of the experiments for ND, OSR, and OOD settings under different training modes.

(a) ND

| Method | Training Mode | Attack | Dataset | | | | | |
|---|---|---|---|---|---|---|---|---|
| | | | CIFAR10 | CIFAR100 | MNIST | FashionMNIST | Head-CT | Covid19 |
| Ours | Non-Adversarial | Clean / PGD-100 | 93.1 / 0.0 | 86.6 / 0.0 | 98.4 / 0.0 | 94.8 / 0.0 | 96.1 / 0.0 | 89.2 / 0.0 |
| | Adversarial | Clean / PGD-100 | 87.4 / 71.1 | 79.6 / 62.8 | 99.4 / 95.7 | 95.6 / 88.1 | 87.3 / 70.0 | 79.6 / 59.4 |

(b) OSR

| Method | Training Mode | Attack | Dataset | | | |
|---|---|---|---|---|---|---|
| | | | CIFAR10 | CIFAR100 | MNIST | FashionMNIST |
| Ours | Non-Adversarial | Clean / PGD-100 | 84.3 / 0.0 | 69.0 / 0.0 | 99.1 / 0.0 | 91.9 / 0.0 |
| | Adversarial | Clean / PGD-100 | 79.6 / 48.5 | 64.1 / 37.7 | 97.2 / 88.8 | 87.7 / 67.1 |

(c) OOD

| Method | Training Mode | Attack | Dataset | |
|---|---|---|---|---|
| | | | CIFAR10 vs CIFAR100 | CIFAR100 vs CIFAR10 |
| Ours | Non-Adversarial | Clean / PGD-100 | 83.0 / 0.0 | 71.2 / 0.0 |
| | Adversarial | Clean / PGD-100 | 75.6 / 38.7 | 61.5 / 30.7 |

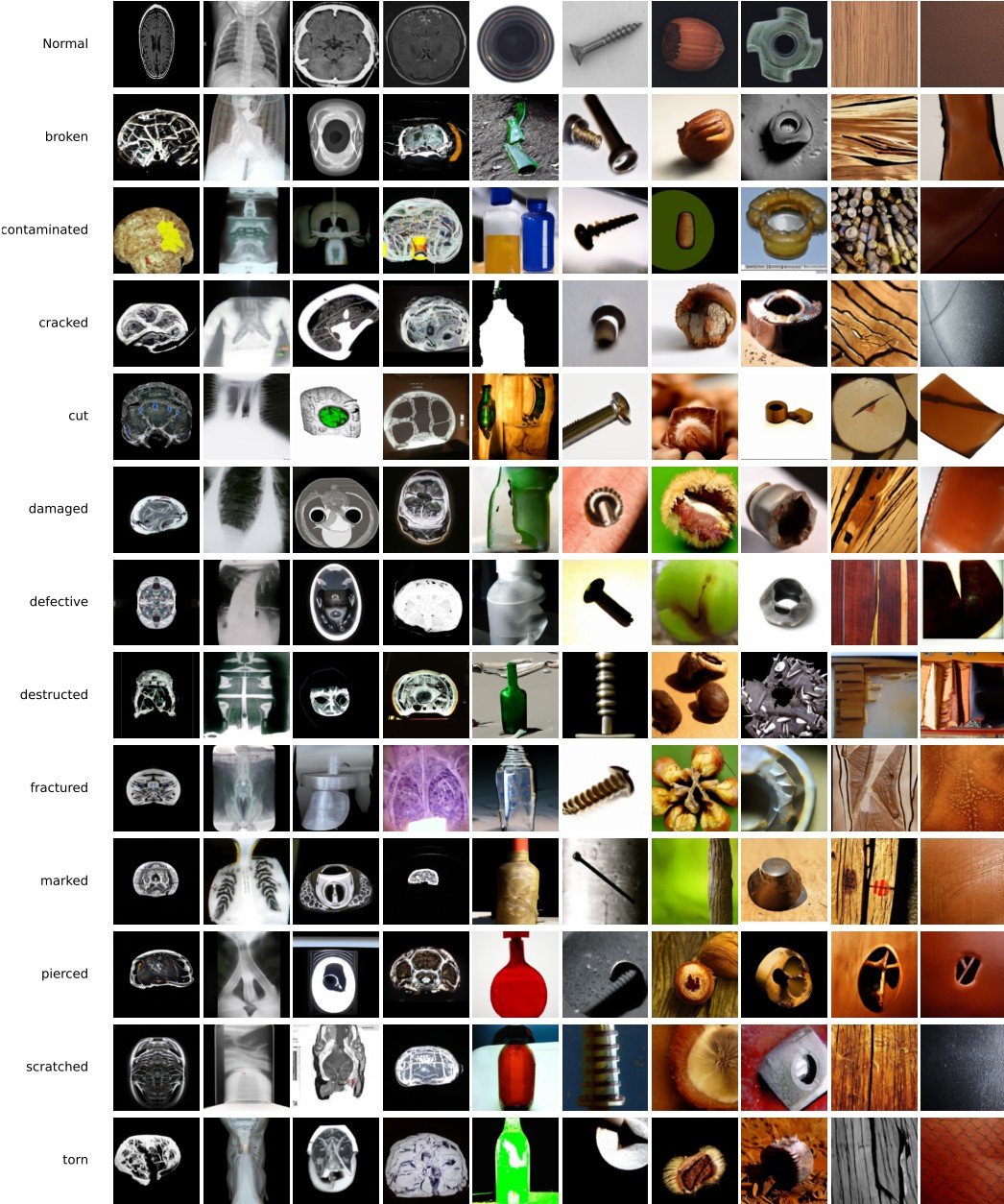

Figure 8: Examples of text-conditioned generation (excluding image conditioning) produced using our pipeline, showcasing the importance of simultaneous image and text conditioning in generating near OOD data.

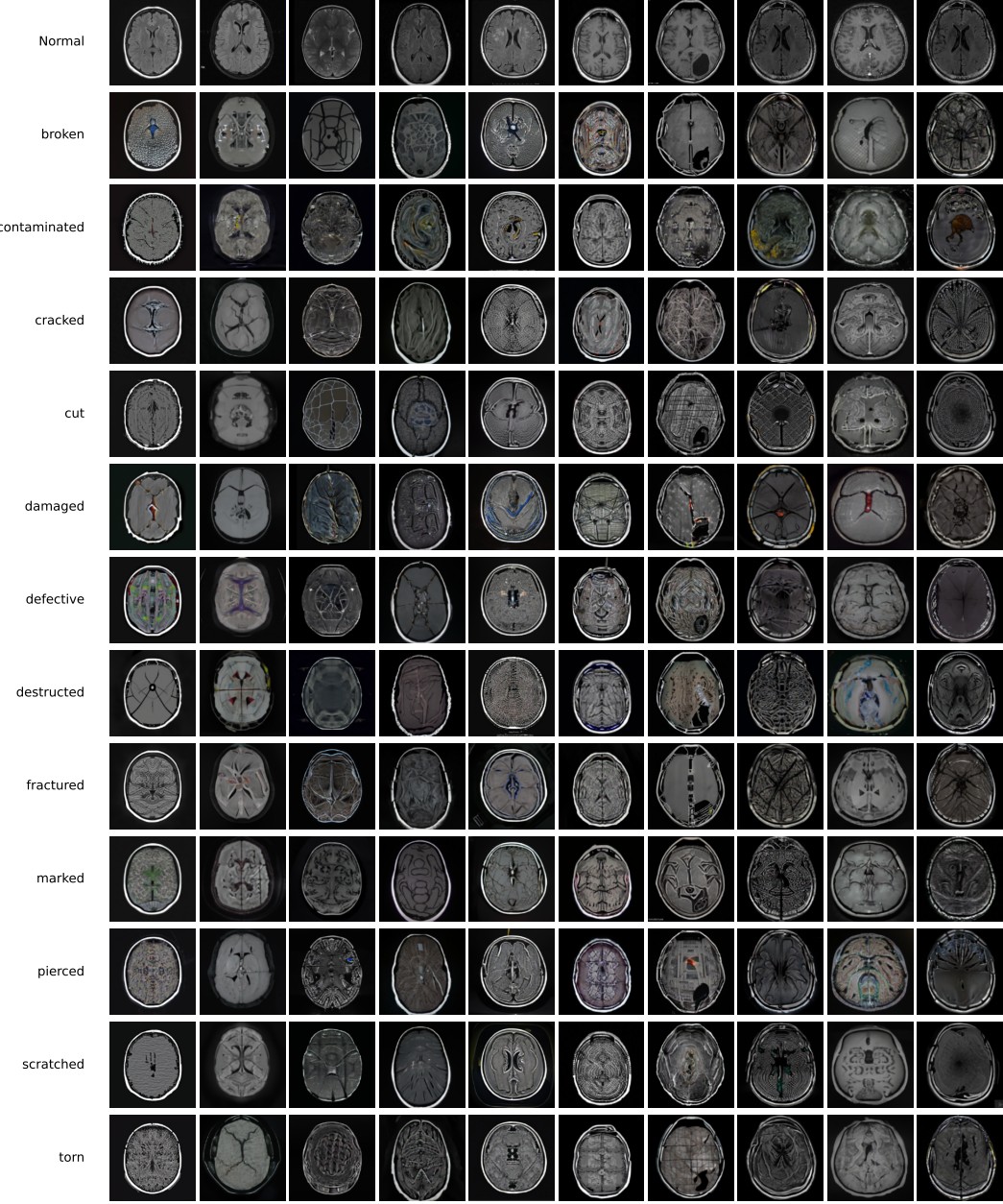

Figure 9: BrainMRI: Examples of generated auxiliary outliers on Dataset BrainMRI Conditioned on Negative Adjectives and inlier Images. The first row depicts inlier images, while subsequent rows demonstrate generated auxiliary outliers corresponding to the negative adjectives written on the left of each row.

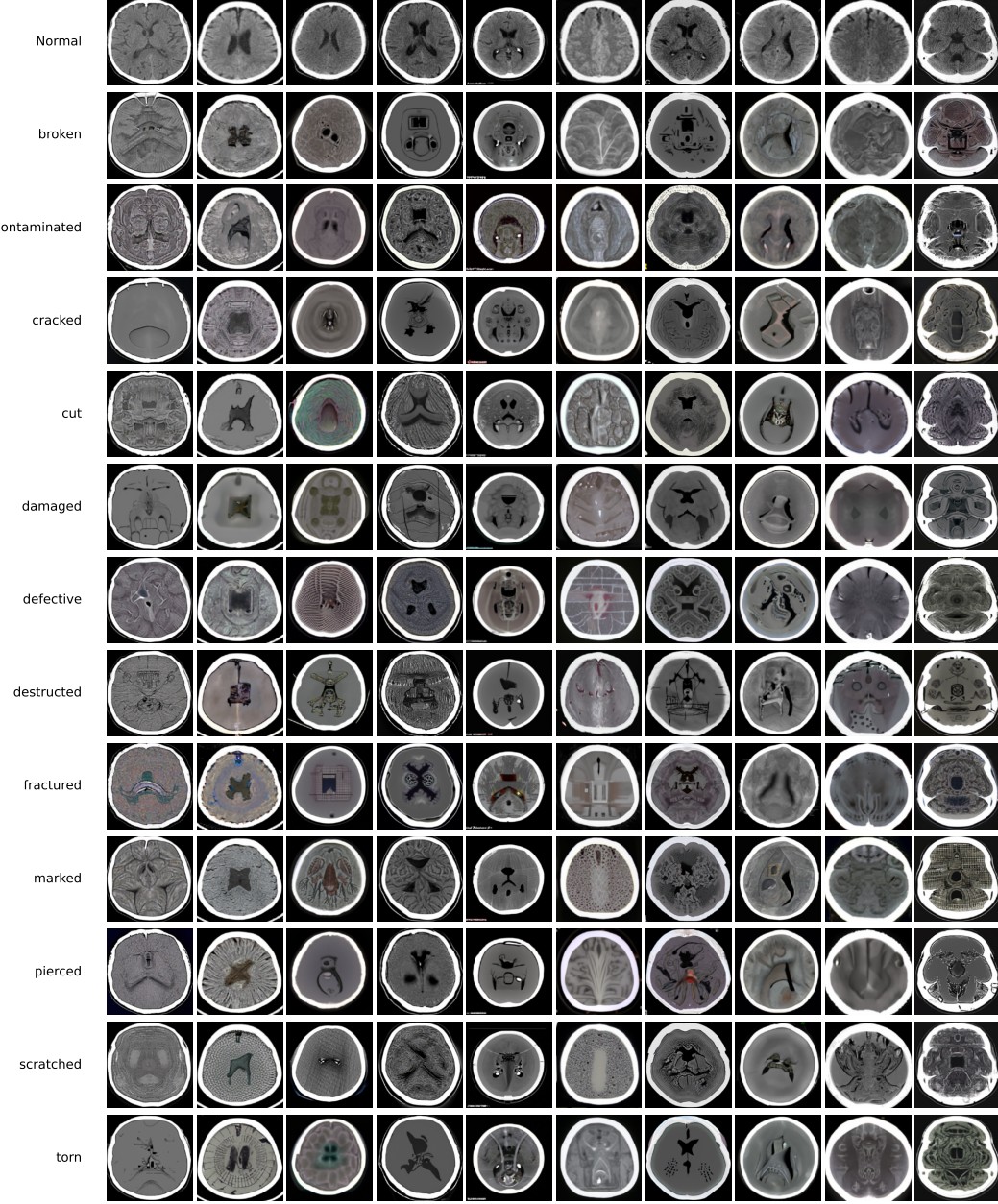

Figure 10: Head-CT: Examples of generated auxiliary outliers on Dataset Head-CT Conditioned on Negative Adjectives and inlier Images. The first row depicts inlier images, while subsequent rows demonstrate generated auxiliary outliers corresponding to the negative adjectives written on the left of each row.

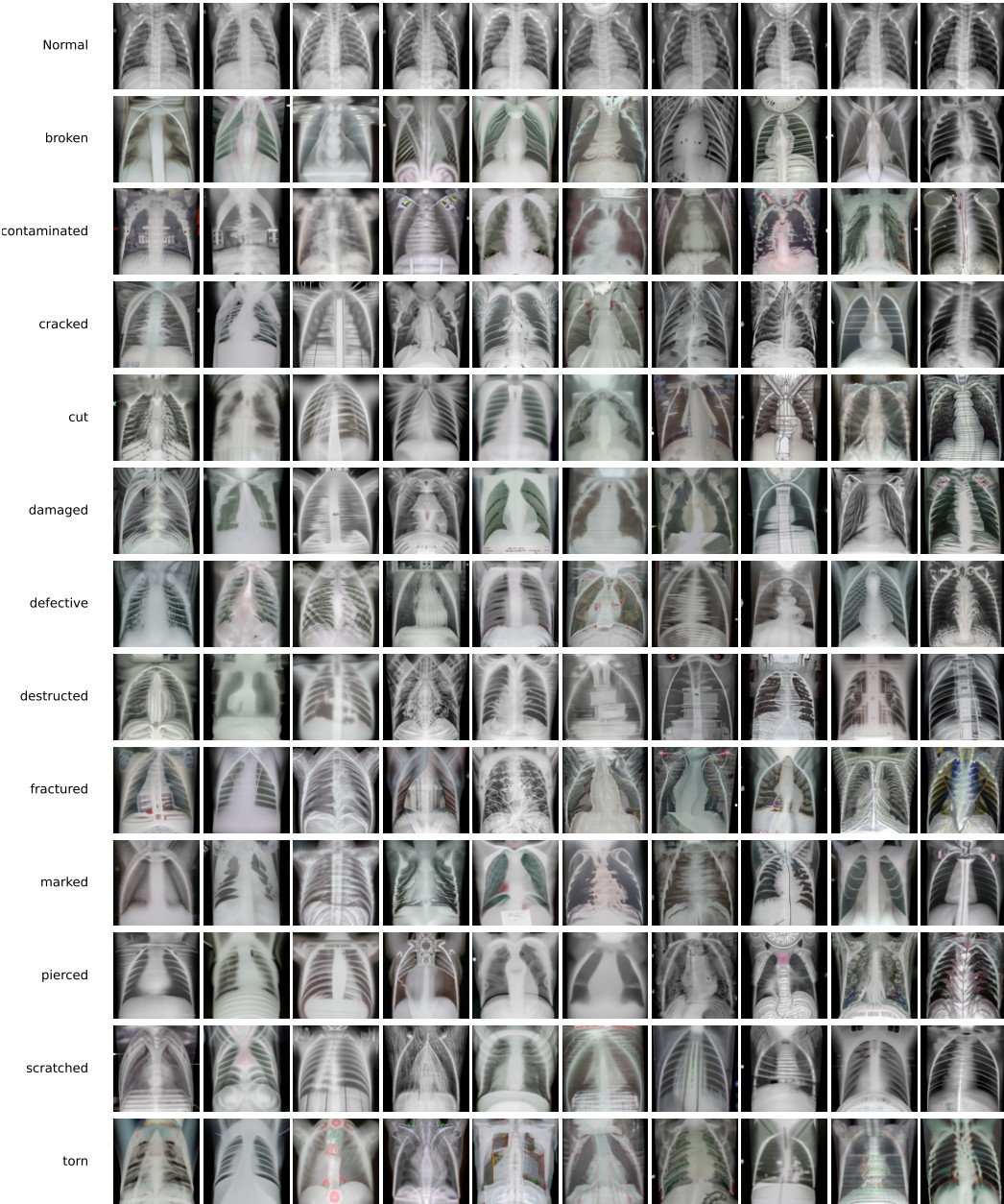

Figure 11: Covid19: Examples of generated auxiliary outliers on Dataset Covid19 Conditioned on Negative Adjectives and inlier Images. The first row depicts inlier images, while subsequent rows demonstrate generated auxiliary outliers corresponding to the negative adjectives written on the left of each row.

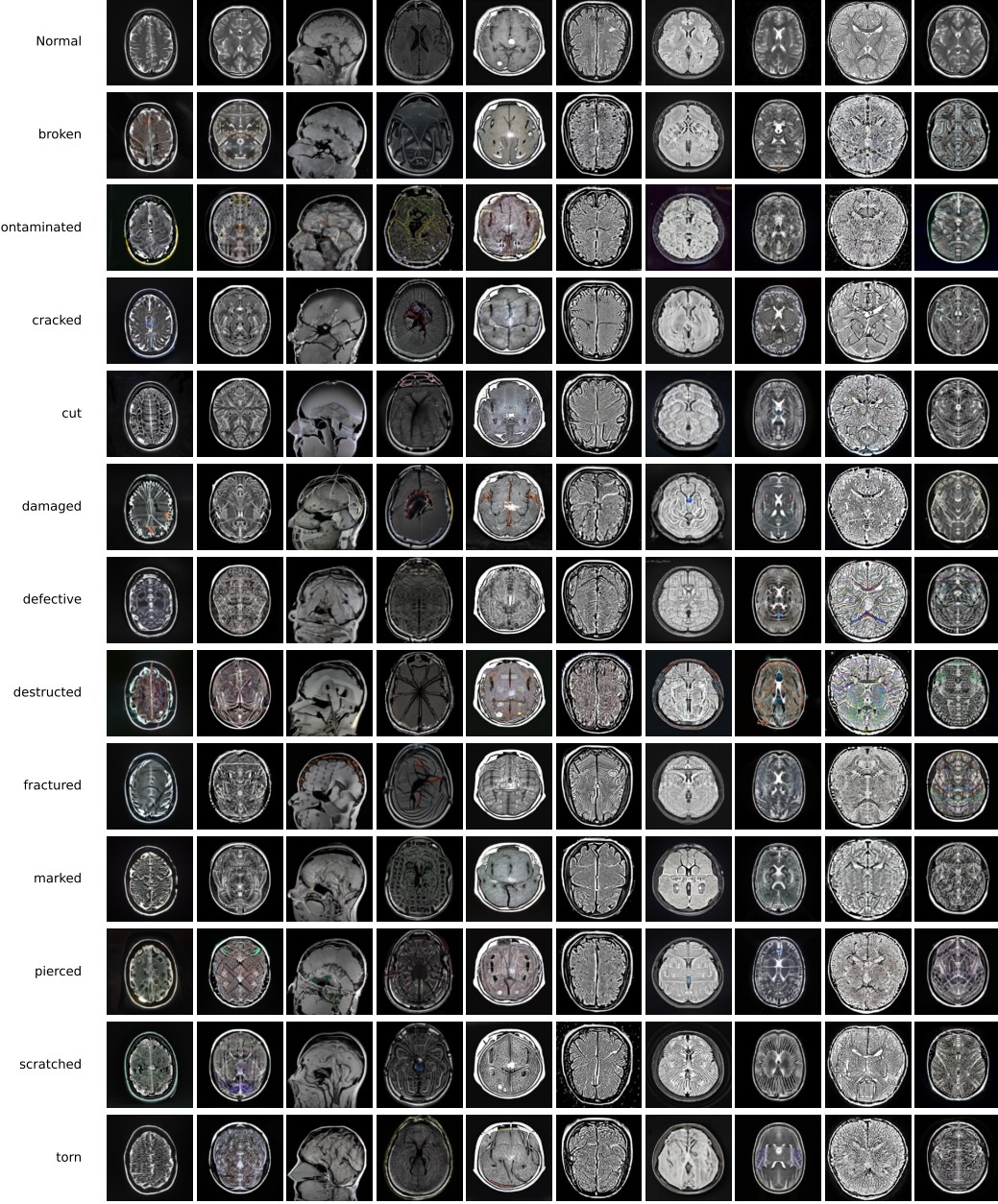

Figure 12: Tumor Detection: Examples of generated auxiliary outliers on Tumor Detection Conditioned on Negative Adjectives and inlier Images. The first row depicts inlier images, while subsequent rows demonstrate generated auxiliary outliers corresponding to the negative adjectives written on the left of each row.

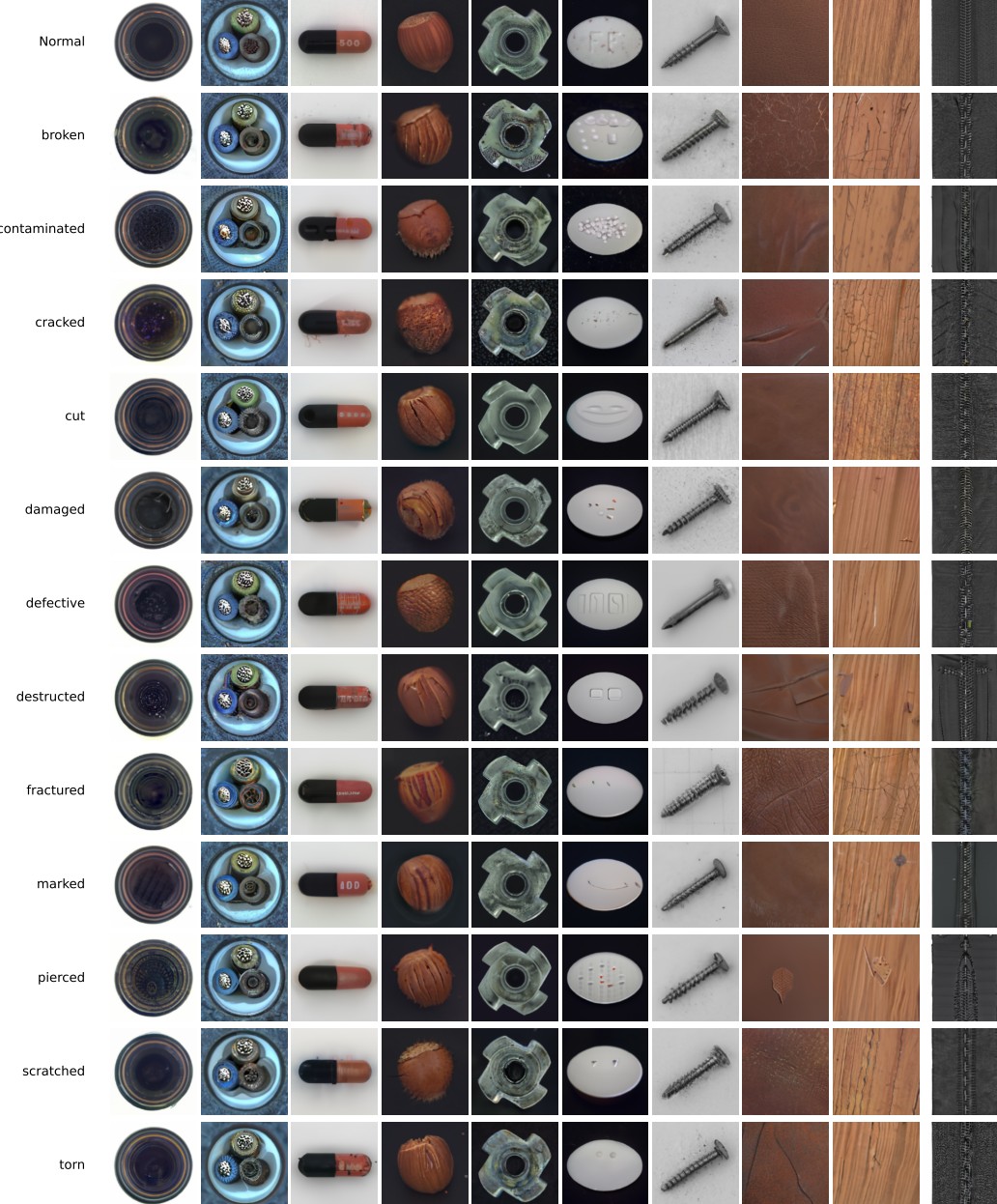

Figure 13: MVTec-AD: Examples of generated auxiliary outliers on Dataset MVTec-AD Conditioned on Negative Adjectives and inlier Images. The first row depicts inlier images, while subsequent rows demonstrate generated auxiliary outliers corresponding to the negative adjectives written on the left of each row.

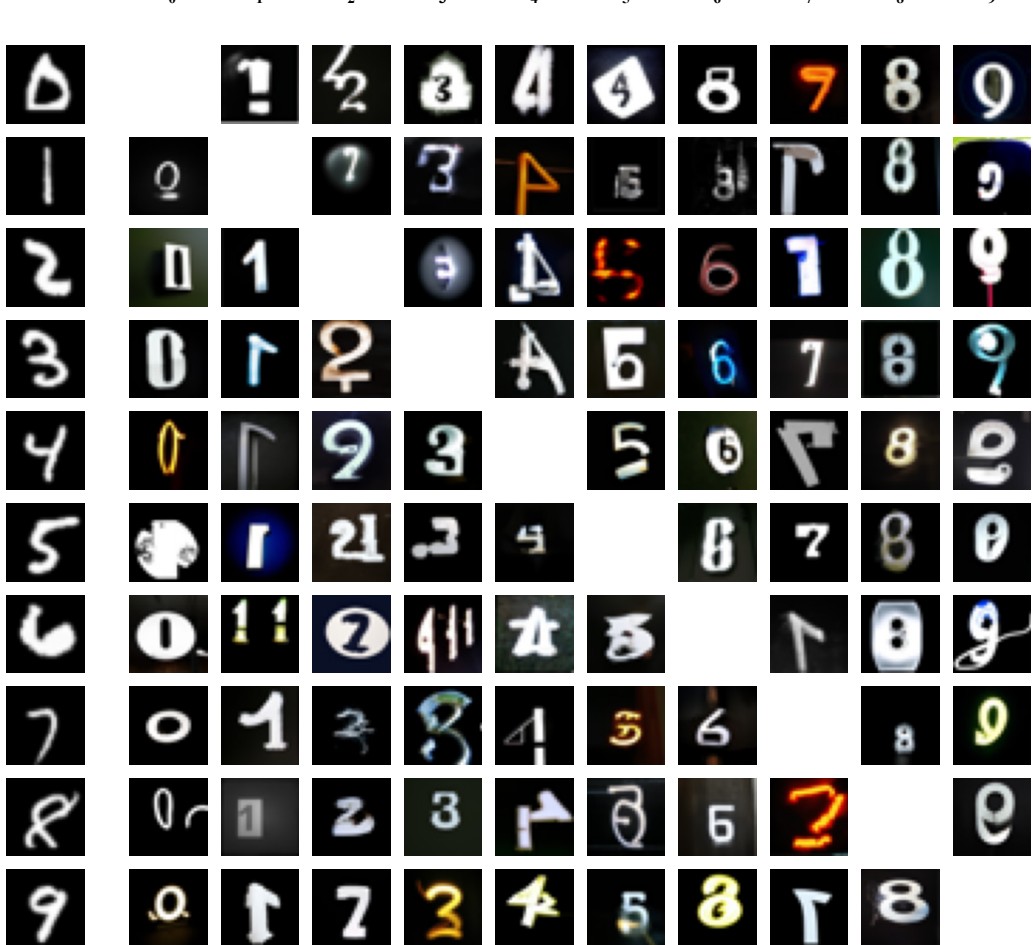

Figure 14: MNIST Adaptive Exposures Grid: This figure illustrates a grid of adaptive exposures of handwritten digits created using the our pipeline with the MNIST dataset, accompanied by their corresponding labels. By utilizing the original data and text prompts, our pipeline generates a variety of exposures that adaptively capture the distribution of the dataset while incorporating outlier elements.

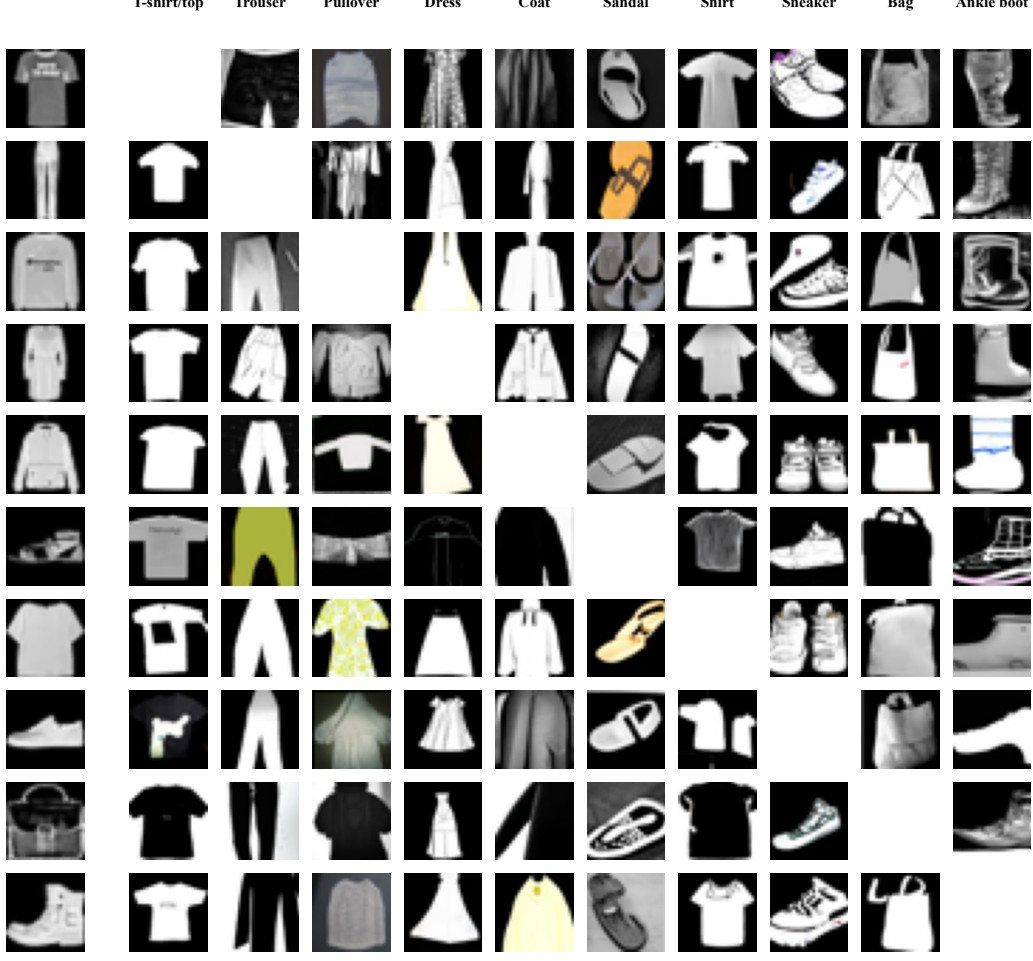

Figure 15: FashionMNIST Adaptive Exposures Grid: This figure showcases a grid of adaptive exposures of fashion items produced by our pipeline with the FashionMNIST dataset, along with their corresponding labels. By leveraging the original data and text prompts, our pipeline generates a range of diverse exposures that closely align with the distribution of the dataset while incorporating outlier elements.

