# OpenReview forum: "RODEO: Robust Out-of-Distribution Detection Via Exposing Adaptive Outliers"
_ICLR.cc/2024/Conference — Submitted to ICLR 2024_

### Official Review · Reviewer_MBna · 2023-10-22

**Soundness:** 2 fair
**Presentation:** 2 fair
**Contribution:** 2 fair
**Rating:** 6
**Confidence:** 5

**Summary:**

This paper studies the challenging problem -- out-of-distribution detection under adversarial attacks. It proposes a novel method RODEO to improve the quality of outlier exposures by generating diverse and near-distribution OOD synthetic samples through leveraging text descriptions of the potential OOD concepts and guiding diffusion model using these texts. The synthetic samples are utilized as OE during the adversarial training of a discriminator. It performs extensive experiments to demonstrate that the proposed method outperforms existing methods, particularly excelling in the highly challenging task of Robust Novelty Detection. The proposed method maintains high OOD detection performance in both standard and adversarial settings across various detection scenarios, including medical and industrial datasets, demonstrating its high applicability in real-world contexts.

**Strengths:**

I think this paper has the following strengths:

1. The idea of using a pre-trained generative diffusion model and combining it with a CLIP model to generate near-distribution outlier data is novel. The results also show that it is effective in improving robust OOD detection performance.

2. The proposed method RODEO achieves significant results on various datasets, including medical and tiny datasets, highlighting the applicability of the work to real-world applications.

3. The proposed RODEO is adaptable to various outlier detection setups, such as ND, OSR, and OOD detection. The method achieves competitive results in clean settings and establishes a SOTA performance in adversarial settings, surpassing existing methods by up to 50% in terms of AUROC.

**Weaknesses:**

I think this paper has the following weaknesses:

1. The setup for theoretical analysis is too simple. It assumes the normal class follows $U(0, a-\epsilon)$ and the anomaly class adheres to $U(a+\epsilon, b)$. However, in practical datasets, the normal class distribution and the anomaly class distribution are much more complicated. It is unclear whether the conclusions drawn from the simple setup still hold for other distributions (or practical datasets). I think the authors can first consider Gaussian distributions to see if the analysis and conclusions still hold.

2. The comparisons with the baselines may not be fair. The proposed method uses a generative diffusion model and a CLIP model, which are pre-trained on millions of data. However, some of the baselines (e.g., ATOM and ALOE) considered in the experiments don't use such extra data for training. Thus, the comparisons may not be fair. It is known that using extra data can significantly improve the model's performance.

3. The PGD attack used is weak. The authors should use a PGD attack with multiple random restarts (e.g., 10 restarts) and more attack iterations (e.g., 1000 steps). Although it uses the strong attack AutoAttack, it doesn't explain how it adapts the AutoAttack to its settings. Since AutoAttack is not designed to attack OOD detectors, it cannot be directly used to attack OOD detectors. Besides, in Tables 2 and 3, the results for AutoAttack are missing.

**Questions:**

1. Could the authors explain whether the theoretical analysis holds for more complex data distributions?

2. Could the authors explain whether the comparisons with the baselines are fair or not?

3. Could the authors explain how they use AutoAttack for evaluation?

---

> ### Author Response · Authors · 2023-11-15
> **Response**
>
> Thank you for taking the time to review our paper and provide your valuable comments. We have conducted experiments and provided results to address your questions. We have also revised our paper with these additional results. Please find our  response to your comments below:
> > **W1&Q1:**
>
> We respectfully request the reviewer to consider our "Common Response" to this question.
>
> > **W2&Q2:**
>
> We understand  the reviewer's concerns regarding the fairness of comparisons due to the use of large pre-trained model. To address this, we would like to provide some key points that highlight our approach's novelty and justify the fairness of our comparisons:
>
> * **Motivation:** Developing robust OOD detection methods is a crucial task due to their application in safety-critical real-world problems such as autonomous driving. While the clean results on challenging benchmarks are near 100% AUROC, the best performance of previous works under attacks is less than random detection even on tiny datasets.
>
> * **Robust Methods:**  Previous robust OOD detection methods mostly focused on a specific setup of outlier detection, such as multi-class outlier detection. For instance, ALOE, ATD, and ATOM are only applicable in the multi-class setup and cannot handle other outlier detection setups like one-class outlier detection. In contrast, our method can generalize across different detection setups and datasets, including medical imaging and industrial datasets, despite the limitations of previous works.
>
> * **RODEO vs. SOTA:** Several studies have utilized pre-trained models to enhance outlier detection. These studies have employed models such as ViTs (trained on 21M samples), BERT (1B data), and CLIP (400M data)[1,2] in the outlier detection domain. It is worth noting that we utilized a smaller version of CLIP, trained on 65M data, which is less data-intensive than the previously mentioned models. Moreover, while previous works directly leveraged the rich features of CLIP to develop a detector, we have adopted a different approach by implicitly using CLIP's guidance.
>
> Furthermore, we conducted an experiment to compare our method with EXOE[2], a recent  SOTA method based on CLIP-400M.  The results indicate that our method slightly outperforms EXOE in clean performance, albeit with a minor margin. On the other hand, the significant superiority of our robust detection method highlights our contribution.
>
> > **W3&Q3:**
>
> OOD detection can be  formulated as:
>
> $$
> g_\lambda(\mathrm{x}) =
> \begin{cases}
> \text{ID} & \text{if } O(\mathrm{x}) \leq \lambda \\\\
> \text{OOD} & \text{if } O(\mathrm{x}) > \lambda
> \end{cases}
> $$
>
> where $O(x)$ is the detection score, and $\lambda$ is the threshold. In PGD attacks, as stated in the Appendix, instead of maximizing the loss value, we try to increase $O(x)$ if $x$ belongs to in-distribution samples and decrease it otherwise. The formulation of the attack would be:
>
> $${x_{0}^*}=x,
> x^{t+1} = x^t + \alpha\cdot \text{sgn}( \nabla_{ x}( y.O_{\boldsymbol{\theta}}(x^t)) )$$
>
> Where $y=1$ for in-distribution samples and $y=-1$ for OOD samples. The same setting holds for AutoAttack. AutoAttack is an ensemble of six attack methods: APGD with CE loss, APGD with DLR loss, APGDT , FAB, multi-targeted FAB, and Square Attack. Due to the presumption in DLR loss-based attacks that the model's output size must be three or greater, we had to exclude these specific attacks in our adaptation of AutoAttack, resulting in a total of four attacks. Addressing your concerns regarding the strength of the PGD attack, we conducted additional experiments with a PGD attack with 1000 steps and 10 random restarts, with results indicating negligible performance reduction. To further validate our method's adversarial robustness, we employed the Adaptive Auto Attack(A3)[3], which has been shown to be a stronger attack than AutoAttack. These enhancements substantially strengthen our novelty detection method's robustness evaluation.
>
>
> ND:
> |Attack\Dataset|CIFAR10|CIFAR100|MNIST|FMNIST|SVHN|MVTecAD|Head-CT|BrainMRI|Tumor Detection|Covid19|
> |-|-|-|-|-|-|-|-|-|-|-|
> |Clean|87.4|79.6|99.4|95.6|75.2|61.5|87.3|76.3|89.0|79.6|
> |PGD-100|71.1|62.8|95.7|88.1|35.4|15.9|70.0|71.1|67.5|59.4|
> |PGD-1000|70.2|62.1|94.6|87.2|33.8|14.9|68.6|68.4|67.0|58.3|
> |AutoAttack|69.3|61.0|95.2|87.6|33.2|14.2|68.4|70.5|66.9|58.8|
> |A3|70.5|61.3|94.0|87.0|31.8|13.4|68.1|67.7|65.6|57.6|
>
> OOD Detection:
> |Dataset\Dataset|Clean|PGD-100|PGD-1000|AutoAttack|A3|
> |-|-|-|-|-|-|
> |CIFAR10|93.2|70.4|69.5|69.0|68.8|
> |CIFAR100|88.1|66.4|64.7|65.3|63.2|
>
> OSR:
> |Attack\Dataset|MNIST|FMNIST|CIFAR10|CIFAR100|
> |-|-|-|-|-|
> |Clean|97.2|88.8|79.6|48.5|
> |PGD-100|87.7|67.1|64.1|37.7|
> |PGD-1000|85.0|65.3|62.7|35.3|
> |AutoAttack|86.4|66.8|63.5|36.9|
> |A3|84.1|62.9|63.0|35.4|
>
> [1] S Esmaeilpour  Zero AAAI 2023
>
> [2] P  Liznerski  Exposing  TMLR 2023
>
> [3] Y Liu Practical CoRR  2022
>
> If our responses address your points, we kindly ask for a re-evaluation of your first score to reflect the paper's enhancements.

---

> > ### Comment · Reviewer_MBna · 2023-11-16
> >
> > Thanks for the response! My concerns have been addressed. Thus, I will raise my score to 6.

---

> > > ### Author Response · Authors · 2023-11-16
> > >
> > > Thank you for your feedback! We're glad to hear your concerns were addressed. We appreciate the time and effort you've put into reevaluating our work.

---

### Official Review · Reviewer_wbuR · 2023-10-22

**Soundness:** 3 good
**Presentation:** 3 good
**Contribution:** 3 good
**Rating:** 8
**Confidence:** 4

**Summary:**

- Detecting out-of-distribution (OOD) inputs during inference is critical for model reliability. However, OOD detection performance degrades significantly under adversarial attacks.
- The paper proposes a data-centric approach called RODEO to improve adversarial robustness of OOD detection by generating effective outliers.
- Key ideas:
(1) Adversarial OOD detection benefits from outliers that are diverse, near the in-distribution data, and conceptually distinct from inliers.
(2) Propose adaptive outlier generation method incorporating text and image information to satisfy above criteria. Uses text encoder to find near-OOD words, CLIP model to guide image generation, and filtering.
(3) Adversarially train classifier on inliers and generated outliers. Use classifier's OOD logit as anomaly score at test time.
- Experiments across novelty detection, open-set recognition and OOD detection show RODEO outperforms under adversarial settings.

**Strengths:**

Originality:
- The concept of generating "near-distribution" outliers that are diverse and conceptually different from the inliers is a unique contribution to the field.
- Creative use of CLIP.

Quality:
- Comprehensive experiments across novelty detection, open-set recognition and OOD detection highlighting broad applicability.
- Strong results surpassing prior work by good margins, especially under adversarial attacks.
- Ablation studies analyzing impact of different outlier generation techniques.
- The paper also provides a detailed theoretical justification for the use of near-distribution outliers.

Clarity:
- Clearly explains and motivates the need for adaptive outlier generation for robustness.

Significance:
- The paper addresses an important problem in machine learning, which is the detection of OOD samples, particularly under adversarial conditions. The proposed method shows significant improvements over existing methods in various detection setups, making it potentially valuable in applications that require robust OOD detection.

**Weaknesses:**

The paper could benefit from a more thorough comparison with other methods of OOD detection, especially those that also use a data-centric approach or those that have been designed specifically for adversarial settings. This could help to place the proposed method in a broader context and demonstrate its advantages and disadvantages more clearly.

The method involves several complex steps, including the generation of outliers and their incorporation into the training process, which might be computationally expensive. The paper has not yet provide a discussion on the trade-off between the performance gain and the added computational cost.

The paper could provide more details on the performance of the method in non-adversarial settings, and it could compare the method with a wider range of existing methods.

**Questions:**

- The paper assumes that effective outliers should be 'near-distribution', 'diverse', and 'conceptually differentiable' from the inliers. While this makes sense in theory, it may not hold in all practical scenarios. There could be situations where effective outliers do not meet all these criteria, and it would be interesting to see how the proposed method would perform in such cases.

- The approach is only evaluated on computer vision tasks and datasets. Would be interesting to see applicability to other modalities.

---

> ### Author Response · Authors · 2023-11-15
> **Response**
>
> We thank the reviewer for the positive review! Thank you for the thoughtful review and valuable comments. Responses to specific points are provided below:
> > **The paper could benefit from a more thorough...**
>
> It should be noted that many of the advanced methods developed for robust detection tasks, including ATD, ALOE, and ATOM, are founded on a data-centric approach. These methods typically employ an outlier exposure (OE) technique, which incorporates an auxiliary random outlier set, like ImageNet, during the process of adversarial training. Our method has been benchmarked against these in Tables 2 and 3 of the paper, and for your convenience, which are provided here as well for your review:
>
> Results on OSR setup:
>
> |Dataset|Mode|ATOM|ALOE|ATD|OURS|
> |-|-|-|-|-|-|
> |MNIST|Clean/PGD|74.8/6.3|79.5/38.2|68.7/56.7|**97.2**/**88.8**|
> |FMNIST|Clean/PGD|64.3/4.7|72.6/29.0|59.6/43.0|**87.7**/**67.1**|
> |CIFAR10|Clean/PGD|68.3/5.2|52.4/25.7|49.0/33.6|**79.6**/**48.5**|
> |CIFAR100|Clean/PGD|51.4/3.2|49.8/18.6|50.5/36.6|**64.1**/**37.7**|
>
> Results on OOD detection setup:
>
> |In-Dataset|Attack|ATOM|ALOE|ATD|OURS|
> |-|-|-|-|-|-|
> |CIFAR10|Clean/PGD|82.7/25.1|**97.8**/6.0|94.3/69.3|93.2/ **70.4**|
> |CIFAR100|Clean/PGD|**91.6**/5.4|79.3/26.4|87.7/55.3|88.1/**66.4**|
>
> > **The method involves several complex steps...**
>
> We have provided details on computational cost in Appendix 10.3. To address your concern, we have added further details about the trade-off between performance gain and the additional computational cost here and have updated the appendix accordingly.
>
> Computational cost:
>
> |Phase|ND|OSR|OOD|
> |-|-|-|-|
> |Adaptive Generation| 30 min|3h| 5h|
> |Adversarial Training|100 min|9h|16h|
>
> To mitigate the computational costs, one alternative is to employ the ImageNet dataset for OE in adversarial training. We have examined this approach in our ablation study. Below, we present the average performance of our detector across two scenarios: (1) using ImageNet as OE for adversarial training, and (2) generating OE with our proposed pipeline and then leveraging it for adversarial training:
>
> |Technique|Clean|Adv|
> |-|-|-|
> |Vanilla OE|75.6| 34.6|
> |Adaptive OE|84.0|66.8|
>
> > **The paper could provide...**
>
> In response to your comment, we have conducted further experiments and included the results here and in the appendix. This includes scenarios where our detector was trained in a non-adversarial setup, with evaluation under both clean and adversarial settings
>
> ND-setup:
>
> |Train Mode|Test Mode|CIFAR10|CIFAR100|MNIST|FMNIST|Head-CT|Covid19|
> |-|-|-|-|-|-|-|-|
> |Clean|Clean|93.1|86.6|98.4|94.8|96.1|89.2|
> ||Adv|0.0|0.0|0.0|0.0|0.0|0.0|
> |||||||||
> |Adv|Clean|87.4|79.6|99.4|95.6|87.3|79.6|
> ||Adv|71.1|62.8|95.7|88.1|70.0|59.4|
>
> OSR-setup:
>
> |Train Mode|Test Mode|CIFAR10|CIFAR100|MNIST|FMNIST|
> |-|-|-|-|-|-|
> |Clean|Clean|84.3|69.0|99.1|91.9|
> ||Adv|0.0|0.0|0.0|0.0|
> |||||||
> |Adv|Clean|79.6|64.1|97.2|87.7|
> ||Adv|48.5|37.7|88.8|67.1|
>
> OOD-setup:
>
> |Train Mode|Test Mode|CIFAR10 vs CIFAR100|CIFAR100 vs CIFAR10|
> |-|-|-|-|
> |Clean|Clean|83.0|71.2|
> ||Adv|0.0|0.0|
> |||||
> |Adv|Clean|75.6|61.5|
> ||Adv|38.7|30.7|
>
>  > **The paper assumes that ...**
>
> We address these considerations in our ablation study, which examines scenarios where auxiliary outlier datasets are either distant from the inlier distribution or lack diversity. The results, extracted from Table 3, help clarify these situations.
>
> |OE\Target|CIFAR10|MNIST|
> |-|-|-|
>  Gaussian Noise|54.4 / 11.3|56.1 / 12.4|
> |Vanilla OE|87.3 / 70.0|90.0 / 43.0|
> |Adaptive OE|**87.4** / **71.1**|**99.4** / **95.7**|
>
> Regarding Near-Distribution, the results show that both Adaptive OE and Vanilla OE (ImageNet) enhance clean and robust detection for the CIFAR10 dataset. However, for MNIST, Vanilla OE (ImageNet) is much less effective compared to our Adaptive method. This difference is likely due to the closer distribution between ImageNet and CIFAR10, in contrast to the larger gap between ImageNet and MNIST distributions.
> Regarding diversity, our results suggest that using Gaussian noise as OE is less effective due to its lower diversity compared to other techniques we evaluated.
> With respect to FDC metric, which serves to assess the diversity and how close is the OE dataset to in-distribution, it is evident that Adaptive OE outperforms the other two methods in this respect. The Ablation Studies section contains tables that compare the AUROC and FDC metrics of our approach with all the OE methods evaluated in our research, and the respective values have been presented here as well:
>
> |OE\Target|CIFAR10|MNIST|
> |-|-|-|
>  Gaussian Noise|0.822|0.704|
> |Vanilla OE|7.647|0.795|
> |Adaptive OE|**8.504**|**11.395**|
>
> >**The approach is only ...**
>
> Your suggestion indeed underscores an important avenue for future research in applying our approach to diverse modalities.
>
> We deeply appreciate your insights, which have been instrumental in elevating our work; we humbly believe these enhancements might resonate in an uplifted assessment.

---

> > ### Comment · Reviewer_wbuR · 2023-11-22
> > **Thank you for your followup.**
> >
> > I appreciate the author's additional experiments and explanations; my rating remains at 8.

---

> > > ### Author Response · Authors · 2023-11-23
> > >
> > > Thank you once again for your insightful feedback and comments.

---

### Official Review · Reviewer_3m3p · 2023-10-25

**Soundness:** 4 excellent
**Presentation:** 2 fair
**Contribution:** 3 good
**Rating:** 6
**Confidence:** 5

**Summary:**

The authors propose adversarial training on synthetic outliers generated by DDPM in order to build a better ND, OSR, and OOD detection model. The key idea is that synthetic outliers must be diverse and near-distribution. To find those, they use DDPM with CLIP guidance with respect to  the nearest neighbors of a label in Word2Vec space (up to a threshold to avoid picking equivalent labels). To keep outliers visually similar, they initialize the diffusion process from an in-distribution image. In experiments, the authors show that the proposed method outperforms approaches like ALOE and APAE.

**Strengths:**

Originality
========
* The idea of guiding DDPM with CLIP embeddings of concepts that are close to in-distribution labels in order to create outliers is novel and interesting.
* I found the idea of using negative adjectives also interesting.

Quality
=====
* The proposed method is sound in general and builds on existing and proven approaches like ALOE.
* The authors provide simplified (1d) theoretical insights on why their method works.
* Ablation studies are provided.

Clarity
=====
* I liked the effort made by the authors to provide some intuition on why their method works (e.g. t-SNE in Figure 4).

Significance
=========
* The proposed approach achieves significant performance improvement.

**Weaknesses:**

Originality
=======
* Generating images with CLIP guidance could be seen like having an additional dataset (the one used to train CLIP) from where outliers are searched. This could be unfair with respect to approaches that do not have access to additional data.
* In fact, CLIP alone has been shown to be good at OOD detection [A] (not discussed in the submission)

Quality
=====
* The authors do not describe the limitations of their work, e.g. what happens if the OOD class is also OOD for clip and the diffusion model?
* Does the 1d analysis hold in higher dimensions?

Clarity
=====
* What is $c$ in e.g., $\mu(x_t|c)$?
* In the Appendix, $\tau$ is defined differently than in page 6 (equation numbers missing). Is there any reason for that?

Minor
====
* Page 6: Adverserial
* "optimizer and PGD-10": PGD is an optimizer too, which causes confusion
* Page 15: an extra labels.
* "The generation process should incorporate the normal distribution": although this is not a typo, when I read it for first time I understood normal meant $\mathcal{N}$ and not in-distribution.
* Figure 5: purturb

[A] Michels, Felix, et al. "Contrastive Language-Image Pretrained (CLIP) Models are Powerful Out-of-Distribution Detectors." arXiv preprint arXiv:2303.05828 (2023).

**Questions:**

I find this work interesting but there are some issues (see weaknesses) that should be addressed:

* Could discuss the limitations of your work (see originality and quality in weaknesses)?
* Could you improve the clarity of the text (see clarity in weaknesses)?

---

> ### Author Response · Authors · 2023-11-14
> **Response**
>
> We appreciate your insightful review and sincerely apologize for the typos. These issues have been addressed in the revised version, and we will enhance the clarity of our paper. Here is our detailed response:
>
> > **Originality :**
>
> We understand concerns about using large pre-trained models akin to extra datasets. Yet, notable methods like ATD and ALOE also use external datasets like TinyImageNet as OE dataset. To highlight our work's novelty, here are some key points:
>
> * **Motivation :** Developing robust outlier detection methods is essential for safety-sensitive real-world applications, like autonomous driving. Although clean results on challenging benchmarks are near 100% AUROC, the top results from previous works drop to below random levels of detection under attack scenarios, even with tiny datasets.
>
> * **Data Complexity :** We agree with the reviewer that utilizing CLIP may provide supervision for our pipeline, however, the real-world application's demand for reliability necessitates robust methods, even at the expense of data complexity. As the reviewer noted, several studies have utilized pre-trained models to enhance outlier detection. These studies have employed models such as ViTs (trained on 21M samples)[1], BERT (trained on 1B data)[2], and CLIP (trained on more than 400M data)[3,4,5] in the outlier detection domain. Despite their notable performance in clean settings, they are vulnerable to adversarial. In contrast, our approach employs a smaller CLIP variant, trained on 65M samples, which not only withstands strong attacks more effectively but also delivers competitive performance in standard settings.
>
> * **Robust Methods :** Prior robust OOD detection techniques were largely tailored to certain scenarios like multi-class outlier detection, with examples like ALOE, ATD, and ATOM being restricted to this setup. Our method, however, is versatile, effectively working across various detection scenarios and datasets, including those in medical imaging and industry, overcoming the constraints of earlier approaches.
>
> * **Different usage of CLIP :** The utilization of CLIP in previous OOD detection methods differs from our approach. Previous works directly leveraged the rich features of CLIP to develop a detector. However, we implicitly utilize CLIP's guidance to generate near and diverse auxiliary outliers for utilizing adversarial training in subsequent steps.
>
> In order to better address the reviewer's concern, we also compare our method with EXOE[5], a recent state-of-the-art novelty detection method based on a CLIP model trained on 400M data, and provide the results here for review:
>
> |||CIFAR10|CIFAR100|MNIST|FMNIST|SVHN|MVTecAD|Mean|
> |-|-|-|-|-|-|-|-|-|
> |RODEO|CLEAN/PGD-100|87.4/71.1|79.6/ 62.8|99.4/ 95.7|95.6/88.1|78.6/45.4|61.5/ 15.9|83.6/**64.1**|
> |EXOE|CLEAN/PGD-100|99.6/0.6|97.8/0.0|96.0/0.1|94.7/2.4|68.2/0.0|76.2/0.4|**88.7** /0.6|
>
> Our method demonstrates superior effectiveness in outlier detection, outperforming other SOTA techniques in both clean and adversarial environments, with the latter being equally crucial.
>
> > **Quality-1 :**
>
> One limitation of our method is its adversarial performance in fine-grained, pixel-level outlier detection, specifically in datasets like MVTecAD. Although our robust detection outperforms other methods, it still requires further development to achieve promising results. A potential reason for our method's limitation with this dataset is that inliers and outliers have very subtle differences at the texture level, making them susceptible to conversion between inliers and outliers by perturbations from strong attacks like PGD.
>
> > **Quality-2:**
>
> We kindly request that you review our common response.
>
> > **Clarity :**
>
> * The generated outliers must be conceptually distinct from inlier data to prevent misleading our detector during training. Therefore, we have defined two filtering steps using two different thresholds,  \$\\mathcal{T}\_{\\text{Image}}\$
> and
> \$\\mathcal{T}\_{\\text{Label}}\$, each specified distinctly. After extracting Near-OOD labels with word2vec, we initially filter out labels that exhibit significant similarity to the inlier labels, ensuring we do not generate inlier-like data in subsequent steps. A second threshold is applied after generating images with the CLIP's guidance, targeting the removal of images that are highly similar to inliers, thus excluding those that could be misconstrued as inlier. This strategy directly supports our main claim that a fundamental aspect of OE is its conceptual differentiation from the inlier.
>
> * This was a typo; 'C' must be replaced with $y_{\text{NOOD}}$.
>
> [1]M Cohen Transformaly CVPR 2023
>
> [2]K XU  Unsupervised ACL 2021
>
> [3]S Esmaeilpour Zero AAAI 2023
>
> [4]M Felix Contrastive arXiv 2023
>
> [5]P  Liznerski Exposing TMLR 2023
>
> Should you find that our revisions have successfully addressed your initial concerns, we would be grateful for your consideration in positively adjusting the paper's score.

---

> > ### Comment · Reviewer_3m3p · 2023-11-19
> > **Response to rebuttal**
> >
> > After reading the responses to all reviewers I believe the quality of this work has improved and I am raising the score to 6 accordingly.

---

> > > ### Author Response · Authors · 2023-11-19
> > >
> > > Thank you! We appreciate your time for reviewing our paper and reading our responses.

---

### Official Review · Reviewer_JDt4 · 2023-10-31

**Soundness:** 2 fair
**Presentation:** 2 fair
**Contribution:** 2 fair
**Rating:** 6
**Confidence:** 2

**Summary:**

This paper argues that both near and diverse outliers are useful to enhance robust outlier detection. Next, they propose an adaptive OE method to generate near and diverse outliers by incorporating both text and image domain information. Finally, a series of experiments demonstrates the effectiveness of this method.

**Strengths:**

1.	The experiment in this article is solid. This paper conducts a series of experiments across various detection setups, such as novelty detection (ND), Open-Set Recognition (OSR), and OOD detection.
2.	The idea has some novelty. This paper argues that near outliers can benefit the OOD detection, and generates near and diverse outliers to train OOD detection, which sounds interesting.

**Weaknesses:**

1.	The theoretical insights of this article lack effective support. In the theoretical insights section, the schematic diagram of this article is  intuitive. However, this article lacks a further explanation of the theory, which makes it appear less convincing. I recommend the author to conduct a more detailed theoretical derivation.

2.	The explanation of Figure 1 is confusing. I am confusing about the explanation about “This suggests that an OE dataset closer to the normal dataset distribution is significantly more beneficial than a distant one.” Does this mean that the samples obtained from previous adversarial training are not effective? The author needs a more detailed explanation here.

3.	When the author explained that near and reverse outliers can enhance the model's OOD detection performance, I was curious about the performance of those distance outliers, which is also a critical part in OOD. (e.g., Gaussian Noise claimed in Fig. 1.)

**Questions:**

see the weakness

---

> ### Author Response · Authors · 2023-11-14
> **Response**
>
> Thank you for your review and useful comments. We have made a revision based on the reviewers' concerns, and specific comments are answered below:
> > **W1:**
>
> We kindly refer you to our common response, which provides a more detailed theoretical derivation.
>
> > **W2-1:**
>
> We apologize for any inconvenience caused by the lack of clarity in the caption of Figure 1. We have since revised the caption to provide a clearer understanding and would like to offer additional insight into Figure 1 here.
>
> Figure 1 represents the motivation for pursuing adaptive outlier exposure (OE) in our paper. OE refers to a technique used in outlier detection literature, which leverages a fixed auxiliary outlier dataset (e.g., ImageNet) during training to improve detection methods by providing more information about outlier data.
>
> In the cases shown in Figure 1, CIFAR10 is treated as the inlier training set, while CIFAR100 serves as a source of Out-of-Distribution (OOD) data unavailable during training time. The ALOE and ATD aim to robustly discriminate them during test time. Both methods employ Tiny ImageNet as an auxiliary outlier set (i.e., OE) during adversarial training, as integral components of their proposed frameworks. In our experiment, while keeping all other aspects of the original methods constant, we replaced Tiny ImageNet with SVHN, MNIST, and Gaussian noise and repeated the experiments for both ALOE and ATD. This replacement led to a notable decline in OOD detection performance for ALOE and ATD on the CIFAR10 vs. CIFAR100 task, particularly under adversarial attack conditions. We attribute this performance drop to the fact that the SVHN, MNIST, and Gaussian Noise distributions are more distant from CIFAR10 (the inlier distribution in this task) compared to Tiny ImageNet. The results are also presented in detail here:
>
> |Methods  \ Auxilary Outlier Dataset (OE) |Tiny Imagenet | SVHN |MNIST |Gaussian Noise |
> |-|-|-|-|-|
> | ALOE(Clean/Adv.) |  78.8/17.0 | 61.2/5.3 |58.7/2.3 |48.2/1.1 |
> | ATD (Clean/Adv.) | 83.1/37.4 | 78.5/ 33.8 |47.2/5.0 |38.1/3.0 |
>
> > **W2-2:**
>
> Previous methods utilized random, fixed auxiliary outlier datasets as OE, which we believe make their methods ineffective when the inlier distribution and the auxiliary outlier set are distant. We support this with our Figure 1, theoretical insight, and extensive ablation studies. For instance, as our Table 4 results indicate, using ImageNet as OE for MNIST as the inlier set is less effective compared to using ImageNet as OE for CIFAR10 as the inlier set. This is because both ImageNet and CIFAR10 include natural images and have less distribution distance compared to ImageNet and MNIST.
>
> > **W3:**
>
> Our ablation study incorporates several alternative OE datasets/techniques for comparison with our proposed adaptive OE method. To address your concern regarding the performance of distant outliers in OOD detection, such as Gaussian Noise referenced in Figure 1, we have conducted experiments using Gaussian Noise as OE. We have updated our tables with these results and provide them here for your review:
>
> | Exposure Technique | CIFAR10 | MNIST | FMNIST | MVTec-ad | Head-CT | Covid19 | Mean  |
> |-|-|-|-|-|-|-|-|
> | Gaussian Noise  | 54.4/11.3 | 56.1/12.4 | 52.7/15.7 | 47.9/0.1 | 49.0/0.8 | 50.7/0.0 | 51.8/7.4 |
> | Adaptive OE (Ours)| 87.4/71.1 | 99.4/95.7 | 95.6/88.1 | 61.5/15.9 | 87.3/70.0 | 79.6/59.4 | 84.0/66.8 |
>
> In ablation studies tables, we have updated the tables accordingly by adding Gaussian Noise as an additional OE technique. Here we have brought the updated AUROC and FDC tables respectively:
>
> |Exposure Technique|CIFAR10|MNIST|FMNIST|MVTec-ad|Head-CT|Covid19|Mean|
> |-|-|-|-|-|-|-|-|
> |Gaussian Noise|54.4/11.3|56.1/12.4|52.7/15.7|47.9/0.1|49.0/0.8|50.7/0.0|51.8/7.4|
> |Vanilla OE|87.3/70.0|90.0/43.0|93.0/82.0|**64.6**/0.5|61.8/2.1|62.7/24.5|75.6/34.6|
> |Mixup|59.4/31.5|59.6/1.7|74.2/48.8|58.5/1.4|54.4/21.4|69.2/50.8|62.8/27.6|
> |Fake Image Generation|29.5/16.2|76.0/51.3|52.2/31.1|43.5/7.3|63.7/6.9|42.7/13.0|51.2/22.5|
> |Stable Diffusion Prompt|62.4/35.6|84.3/62.5|63.7/48.5|54.9/12.6|71.5/3.6|37.1/0.0|60.1/23.5|
> |Ours|**87.4**/**71.1**|**99.4**/**95.7**|**95.6**/**88.1**|61.5/**15.9**|**87.3**/**70.0**|**79.6**/**59.4**|**84.0**/**66.8**|
>
> |Exposure Technique|CIFAR10|MNIST|FMNIST|MVTec-ad|Head-CT|Covid19|Mean|
> |-|-|-|-|-|-|-|-|
> |Gaussian Noise|0.822|0.704|0.810|0.793|0.735|0.674|0.756|
> |Vanilla OE|7.647|0.795|2.444|1.858|3.120|3.906|3.295|
> |Mixup|2.185|0.577|0.985|1.866|1.587|2.078|1.547|
> |Fake Image Generation|0.987|0.649|0.907|1.562|0.881|0.896|0.980|
> |Stable Diffusion Prompt|1.359|1.790|1.399|**1.982**|1.097|0.654|1.381|
> |Ours|**8.504**|**11.395**|**3.819**|1.948|**12.016**|**5.965**|**6.687**|
>
> We have strived to address each point you raised comprehensively. If our rebuttal has resolved these issues to your satisfaction, we would greatly appreciate it if you could reassess your scoring of our paper.

---

> > ### Author Response · Authors · 2023-11-21
> > **A Friendly Reminder Regarding the Rebuttal Conclusion**
> >
> > Dear Reviewer JDt4,
> > Thank you once again for your thoughtful comments and constructive suggestions. We have carefully addressed your feedback, providing thorough responses and additional experimental results in our revised manuscript. We would be grateful if you could review our responses and consider updating your evaluation scores if you find that your concerns have been adequately addressed. We remain open to further discussion on any points that you feel have not been fully resolved.
> >
> > Thank you for your time and attention.
> >
> > Best regards, Authors

---

> > > ### Author Response · Authors · 2023-11-23
> > > **A Friendly Reminder for Reviewer JDt4**
> > >
> > > Dear Reviewer JDt4,
> > >
> > > We truly value the insights and recommendations you’ve offered. Pursuant to your comments, we've updated our manuscript and provided individual responses to each of your points in the comments section. We kindly invite you to examine these updates and consider revising your evaluation score should you find your concerns have been effectively resolved.
> > >
> > > Best regards,
> > > Authors

---

### Author Response · Authors · 2023-11-14
**Common Response**

In this response, we provide a more detailed theoretical derivation that highlights the importance of employing near and diverse outlier exposure samples to enhance robust detection.
Let's assume that the normal data is coming from $\mathcal{N}(0, I)$ and the anomaly is distributed according to $\mathcal{N}(a, I)$. Furthermore, let $N(a^\prime, I)$ be the outlier exposure data. We assume that the OE is farther away from the normal class than the anomaly data, i.e. $\| a^\prime \| \geq \| a \|$. Assuming access to a large training set of normal and exposure samples, the optimal classifier would be $y = \frac{a^{\prime\top}}{\|a^\prime\|} (x - \frac{a^\prime}{2}) = \frac{a^{\prime\top}}{\|a^\prime\|} x - \frac{\|a^\prime\|}{2}$, for an adversary that has a budget of at most $\epsilon$ perturbation in $\ell_2$ norm [1]. Now, applying this classifier on the normal and anomaly classes at the test time, we get:
$$
\frac{a^{\prime\top} x}{\|a^\prime\|} \sim \mathcal{N}(0, I),
$$
for a normal $x$, and also:
$$
\frac{a^{\prime\top} x}{\|a^\prime\|} \sim \mathcal{N}(\frac{a^\top a^\prime}{\|a^\prime\|}, I),
$$
for an anomalous $x$. Therefore, using the trained classifier $y$ to discriminate the normal and anomaly classes, the error rate would be:
$$
( 1 - \Phi( \frac{\|a^\prime\|}{2} - \epsilon)) + ( 1 - \Phi(\frac{a^\top a^\prime}{\|a^\prime\|} - \frac{\|a^\prime\|}{2} - \epsilon)),
$$
where $\Phi(.)$ is the CDF for the normal distribution $\mathcal{N}(0, 1)$.

Let $\delta = a^\prime - a$, and note that:
$$
    \frac{a^\top a^\prime}{\|a^\prime\|} - \frac{\|a^\prime\|}{2}  = \frac{(a^\prime - \delta)^\top a^\prime}{\|a^\prime\|} - \frac{\|a^\prime\|}{2}
    = \frac{\|a^\prime\|}{2} - \frac{\delta^\top a^\prime}{\| a^\prime \|}.
$$
But note that:
$$
    \delta^\top a^\prime  = a^{\prime \top} a^\prime - a^\top a^\prime
     =  \| a^{\prime} \|^2 - \| a \| \| a^\prime \| \cos(\theta)
     = \| a^\prime \| ( \| a^\prime \| - \| a \| \cos(\theta))
     \geq \| a^\prime \| ( \| a^\prime \| - \| a \|) \geq 0,
$$
because we have previously assumed $\|a^\prime \| \geq \| a \|$ to reflect that the OE could be far-distribution. Hence, note that the error rate can be written as:
$$
    ( 1 - \Phi\left( \frac{\|a^\prime\|}{2} - \epsilon\right)) + ( 1 - \Phi\left(\frac{\|a^\prime\|}{2} - \frac{\delta^\top a^\prime}{\| a^\prime \|} - \epsilon\right))  = ( 1 - \Phi\left( \frac{\|a^\prime\|}{2} - \epsilon\right)) + ( 1 - \Phi\left(\frac{\|a^\prime\|}{2} - c - \epsilon\right)),
$$
where $c \geq 0$. Note that for a fixed $\| a^\prime \|$, by making  $\cos(\theta)$ small, the error increases, as $\Phi$ is an increasing function. Also, note that for the case that $\theta = 0$, i.e. smallest possible error among fixed $\|a^\prime \|$, the error can be rewritten as:
$$
     ( 1 - \Phi\left( \frac{\|a \|}{2} + \frac{(\|a^\prime \| - \|a \|)}{2} - \epsilon\right)) + ( 1 - \Phi\left(\frac{\|a \|}{2} - \frac{(\| a^\prime \| - \|a \|)}{2} - \epsilon\right))  = ( 1 - \Phi\left( \frac{\|a \|}{2} + d - \epsilon\right)) + ( 1 - \Phi\left(\frac{\|a \|}{2} - d - \epsilon\right)),
$$
with $d = \frac{(\|a^\prime \| - \|a \|)}{2}  \geq 0$. Note that if $d$ is close to zero, i.e. near-distribution OE, the error converges to that of the adversarial Bayes optimal. But as $d$ grows large, the error becomes larger. Therefore, the more OE is away from the normal distribution, the larger the error rate becomes.

Now, let's assume that the OE follows a less *diverse* distribution, i.e. $\mathcal{N}(a^\prime, \sigma^2 I)$, with $\sigma < 1$. In this case, the intercept of the optimal line that separates the two class gets biased towards the OE distribution, increasing the error rate of classifying normal vs. anomaly. Again, to make this error small, one has to increase $\sigma^2$ to a limit that matches the original anomaly distribution $\sigma^2 = 1$.

[1] L. Shmidt et al, ``Adversarially Robust Generalization Requires More Data,'' 2018.

---

### Meta-Review · Area_Chair_fET5 · 2023-12-22

**Metareview:**

The authors suggest a method for adversarially robust anomaly/out of distribution detection. The main contribution is the generation of near outlier image based on text-to-image based image generation using diffusion processes. The generated outliers are modeled as additional class and then the authors do standard adversarial training on top.

Strength:
- the authors report consistent improvements over prior work on adversarially robust anomaly/OOD detection

Weakness:
- the evaluation of the attack on the detector has to be done using a strong (adaptive) attack. In the paper and in the appendix there is now a mix of different techiques. The authors should use AutoAttack or similar frameworks for evaluation everywhere. Additionally, there should be runs with a large number of PGD steps (potentially on a subset of points) and a large number of restarts to ensure that the evaluation is correct. Given that there are runs where PGD with 1000 steps, which are better than AutoAttack, is an indication that robustness is overestimated.

- the main contribution of the paper is the generation of near outliers. However, there is no comparison (nor citation) of a large number of works using generative techniques. In [A] they use a GAN to produce near outliers, so the idea looks very similar to the present paper. [B] uses even outlier exposure together with generated outliers from a GAN. Other techniques involve virtual outlier synthesis in feature space like [C,D]. All these papers need to be discussed and compared to. [E] seems very related but is not take into account as it is concurrent work.

[A] Lee et al, Training Confidence-calibrated Classifiers for Detecting Out-of-Distribution Samples, ICLR 2018.
[B] Kirchheim et al, On outlier exposure with generative models, NeurIPS ML Safety Workshop, 2022
[C] Du et al, Learning what you don’t know by virtual outlier synthesis. ICLR 2022
[D] Tao et al. Non-parametric outlier synthesis. ICLR 2023
[E] Du et al, Dream the Impossible: Outlier Imagination with Diffusion Models, arXiv:2309.13415
 (concurrent work, not taken into account for the decision)

- regarding adversarially robust OOD detection, in [F] they suggest ACET, where they do adversarial training on synthetic OOD data, [G] seems very similar to the present paper where they do adversarial training on in- and out-distribution for a very rich out-distribution. [H] uses IBP to achieve certified adversarially robust OOD detection and reports to outperform or perform similar to ATOM. [I] use diffusion denoised smoothing and get $\ell_2$ and $\ell_\infty$-guarantees and show results for clean and robust OOD detection which seem to better than what is achieved in the present paper. All these papers are not cited and not compared to in the submission.

[F] Hein et al, Why relu networks yield high-confidence predictions far away from the training data and how to mitigate the problem, CVPR 2019
[G] Augustin et al, Adversarial Robustness on In- and Out-Distribution Improves Explainability, ECCV 2020
[H] Meinke et al, Provably Robust Detection of Out-of-distribution Data (almost) for free, NeurIPS 2022
[I] Franco et al, Diffusion Denoised Smoothing for Certified and Adversarial Robust Out-Of-Distribution Detection, AISafety workshop (IJCAI 2023),  arXiv:2303.14961

In total, this papers suggests an interesting way to generate outliers for OE training. However, the authors missed major prior work both in the area of outlier generation as well as adversarially robust OOD detection. Thus this paper is not ready yet for publication, but I encourage the authors to do the necessary comparisons and then resubmit.

**Justification For Why Not Higher Score:**

Comparison to prior work missing

**Justification For Why Not Lower Score:**

N/A

---

### Decision · Program_Chairs · 2024-01-16

Reject